# Gradient-Free Methods for Deterministic and Stochastic Nonsmooth Nonconvex Optimization

**Tianyi Lin**    **Zeyu Zheng**    **Michael I. Jordan**
University of California, Berkeley
{darren_lin,zyzheng}@berkeley.edu, jordan@cs.berkeley.edu

## Abstract

Nonsmooth nonconvex optimization problems broadly emerge in machine learning and business decision making, whereas two core challenges impede the development of efficient solution methods with finite-time convergence guarantee: the lack of computationally tractable optimality criterion and the lack of computationally powerful oracles. The contributions of this paper are two-fold. First, we establish the relationship between the celebrated Goldstein subdifferential [46] and uniform smoothing, thereby providing the basis and intuition for the design of gradient-free methods that guarantee the finite-time convergence to a set of Goldstein stationary points. Second, we propose the gradient-free method (GFM) and stochastic GFM for solving a class of nonsmooth nonconvex optimization problems and prove that both of them can return a $(\delta, \epsilon)$-Goldstein stationary point of a Lipschitz function $f$ at an expected convergence rate at $O(d^{3/2}\delta^{-1}\epsilon^{-4})$ where $d$ is the problem dimension. Two-phase versions of GFM and SGFM are also proposed and proven to achieve improved large-deviation results. Finally, we demonstrate the effectiveness of 2-SGFM on training ReLU neural networks with the MINST dataset.

## 1   Introduction

Many of the recent real-world success stories of machine learning have involved nonconvex optimization formulations, with the design of models and algorithms often being heuristic and intuitive. Thus a gap has arisen between theory and practice. Attempts have been made to fill this gap for different learning methodologies, including the training of multi-layer neural networks [25], orthogonal tensor decomposition [41], M-estimators [63, 64], synchronization and MaxCut [6, 66], smooth semidefinite programming [15], matrix sensing and completion [10, 42], robust principal component analysis (RPCA) [43] and phase retrieval [82, 79, 64]. For an overview of nonconvex optimization formulations and the relevant ML applications, we refer to a recent survey [51].

It is generally intractable to compute an approximate global minimum [69] or to verify whether a point is a local minimum or a high-order saddle point [67]. Fortunately, the notion of *approximate stationary point* gives a reasonable optimality criterion when the objective function $f$ is smooth; the goal here is to find a point $\mathbf{x} \in \mathbb{R}^d$ such that $\|\nabla f(\mathbf{x})\| \leq \epsilon$. Recent years have seen rapid algorithmic development through the lens of nonasymptotic convergence rates to $\epsilon$-stationary points [70, 44, 45, 20, 21, 53]. Another line of work establishes algorithm-independent lower bounds [22, 23, 3, 4].

Relative to its smooth counterpart, the investigation of nonsmooth optimization is relatively scarce, particularly in the nonconvex setting, both in terms of efficient algorithms and finite-time convergence guarantees. Yet, over several decades, nonsmooth nonconvex optimization formulations have found applications in many fields. A typical example is the training multi-layer neural networks with ReLU neurons, for which the piecewise linear activation functions induce nonsmoothness. Another example arises in controlling financial risk for asset portfolios or optimizing customer satisfaction in service systems or supply chain systems. Here, the nonsmoothness arises from the payoffs of financial

36th Conference on Neural Information Processing Systems (NeurIPS 2022).

derivatives and supply chain costs, e.g., options payoffs [38] and supply chain overage/underage costs [78]. These applications make significant demands with respect to computational feasibility, and the design of efficient algorithms for solving nonsmooth nonconvex optimization problems has moved to the fore [65, 30, 28, 85, 12, 31, 80].

The key challenges lie in two aspects: (i) the lack of a computationally tractable optimality criterion, and (ii) the lack of computationally powerful oracles. More specifically, in the classical setting where the function $f$ is Lipschitz, we can define $\epsilon$-stationary points based on the celebrated notion of Clarke stationarity [26]. However, the value of such a criterion has been called into question by Zhang et al. [85], who show that no finite-time algorithm guarantees $\epsilon$-stationarity when $\epsilon$ is less than a constant. Further, the computation of the gradient is impossible for many application problems and we only have access to a noisy function value at each point. This is a common issue in the context of simulation optimization [68, 48]; indeed, the objective function value is often achieved as the output of a black-box or complex simulator, for which the simulator does not have the infrastructure needed to effectively evaluate gradients; see also Ghadimi and Lan [44] and Nesterov and Spokoiny [72] for comments on the lack of gradient evaluation in practice.

**Contribution.** In this paper, we propose and analyze a class of deterministic and stochastic gradient-free methods for nonsmooth nonconvex optimization problems in which we only assume that the function $f$ is Lipschitz. Our contributions can be summarized as follows.

1. We establish a relationship between the Goldstein subdifferential and uniform smoothing via appeal to the hyperplane separation theorem. This result provides the basis for algorithmic design and finite-time convergence analysis of gradient-free methods to $(\delta, \epsilon)$-Goldstein stationary points.

2. We propose and analyze a gradient-free method (GFM) and stochastic GFM for solving a class of nonsmooth nonconvex optimization problems. Both of these methods are guaranteed to return a $(\delta, \epsilon)$-Goldstein stationary point of a Lipschitz function $f : \mathbb{R}^d \mapsto \mathbb{R}$ with an expected convergence rate of $O(d^{3/2}\delta^{-1}\epsilon^{-4})$ where $d \geq 1$ is the problem dimension. Further, we propose the two-phase versions of GFM and SGFM. As our goal is to return a $(\delta, \epsilon)$-Goldstein stationary point with user-specified high probability $1 - \Lambda$, we prove that the two-phase version of GFM and SGFM can improve the dependence from $(1/\Lambda)^4$ to $\log(1/\Lambda)$ in the large-deviation regime.

**Related works.** Our work is related to a line of literature on gradient-based methods for nonsmooth and nonconvex optimization and gradient-free methods for smooth and nonconvex optimization. Due to space limitations, we defer our comments on the former topic to Appendix A. In the context of gradient-free methods, the basic idea is to approximate a full gradient using either a one-point estimator [39] or a two-point estimator [1, 44, 37, 75, 72], where the latter approach achieves a better finite-time convergence guarantee. Despite the meteoric rise of two-point-based gradient-free methods, most of the work is restricted to convex optimization [37, 75, 83] and smooth and nonconvex optimization [72, 44, 61, 62, 24, 52, 49]. For nonsmooth and convex optimization, the best upper bound on the global rate of convergence is $O(d\epsilon^{-2})$ [75] and this matches the lower bound [37]. For smooth and nonconvex optimization, the best global rate of convergence is $O(d\epsilon^{-2})$ [72] and $O(d\epsilon^{-4})$ if we only have access to noisy function value oracles [44]. Additional regularity conditions, e.g., a finite-sum structure, allow us to leverage variance-reduction techniques [62, 24, 52] and the best known result is $O(d^{3/4}\epsilon^{-3})$ [49]. However, none of these gradient-free methods have been developed for nonsmooth nonconvex optimization and the only gradient-free method we are aware of for the nonsmooth is summarized in Nesterov and Spokoiny [72, Section 7].

## 2 Preliminaries and Technical Background

We provide the formal definitions for the class of Lipschitz functions considered in this paper, and the definitions for generalized gradients and the Goldstein subdifferential that lead to optimality conditions in nonsmooth nonconvex optimization.

## 2.1 Function classes

Imposing regularity on functions to be optimized is necessary for obtaining optimization algorithms with finite-time convergence guarantees [71]. In the context of nonsmooth optimization there are two types of regularity conditions: Lipschitz properties of function values and bounds on function values.

We first list several equivalent definitions of Lipschitz continuity. A function $f : \mathbb{R}^d \mapsto \mathbb{R}$ is said to be *L-Lipschitz* if for every $\mathbf{x} \in \mathbb{R}^d$ and the direction $\mathbf{v} \in \mathbb{R}^d$ with $\|\mathbf{v}\| \leq 1$, the directional projection $f_{\mathbf{x},\mathbf{v}}(t) := f(\mathbf{x} + t\mathbf{v})$ defined for $t \in \mathbb{R}$ satisfies

$$|f_{\mathbf{x},\mathbf{v}}(t) - f_{\mathbf{x},\mathbf{v}}(t')| \leq L|t - t'|, \quad \text{for all } t, t' \in \mathbb{R}.$$

Equivalently, $f$ is $L$-Lipschitz if for every $\mathbf{x}, \mathbf{x}' \in \mathbb{R}^d$, we have

$$|f(\mathbf{x}) - f(\mathbf{x}')| \leq L\|\mathbf{x} - \mathbf{x}'\|.$$

Further, the function value bound $f(\mathbf{x}^0) - \inf_{\mathbf{x} \in \mathbb{R}^d} f(\mathbf{x})$ appears in complexity guarantees for smooth and nonconvex optimization problems [71] and is often assumed to be bounded by a positive constant $\Delta > 0$. Note that $\mathbf{x}^0$ is a prespecified point (i.e., an initial point for an algorithm) and we simply fix it for the remainder of this paper. We define the function class which will be considered in this paper.

**Definition 2.1** *Suppose that $\Delta > 0$ and $L > 0$ are both independent of the problem dimension $d \geq 1$. Then, we denote $\mathcal{F}_d(\Delta, L)$ as the set of $L$-Lipschitz functions $f : \mathbb{R}^d \mapsto \mathbb{R}$ with the bounded function value $f(\mathbf{x}^0) - \inf_{\mathbf{x} \in \mathbb{R}^d} f(\mathbf{x}) \leq \Delta$.*

The function class $\mathcal{F}_d(\Delta, L)$ includes Lipschitz functions on $\mathbb{R}^d$ and is thus different from the nonconvex function class considered in the literature [44, 72]. First, we do not impose a smoothness condition on the function $f \in \mathcal{F}_d(\Delta, L)$, in contrast to the nonconvex functions studied in Ghadimi and Lan [44] which are assumed to have Lipschitz gradients. Second, Nesterov and Spokoiny [72, Section 7] presented a complexity bound for a randomized optimization method for minimizing a nonsmooth nonconvex function. However, they did not clarify why the norm of the gradient of the approximate function $f_{\bar{\mu}}$ of the order $\delta$ (we use their notation) serves as a reasonable optimality criterion in nonsmooth nonconvex optimization. They also assume an exact function value oracle, ruling out many interesting application problems in simulation optimization and machine learning.

In contrast, our goal is to propose fast gradient-free methods for nonsmooth nonconvex optimization in the absence of an exact function value oracle. In general, the complexity bound of gradient-free methods will depend on the problem dimension $d \geq 1$ even when we assume that the function to be optimized is convex and smooth [37, 75]. As such, we should consider a function class with a given dimension $d \geq 1$. In particular, we consider a optimality criterion based on the celebrated Goldstein subdifferential [46] and prove that the number of function value oracles required by our deterministic and stochastic gradient-free methods to find a $(\delta, \epsilon)$-Goldstein stationary point of $f \in \mathcal{F}_d(\Delta, L)$ is $O(\text{poly}(d, L, \Delta, 1/\epsilon, 1/\delta))$ when $\delta, \epsilon \in (0, 1)$ are constants (see the definition of Goldstein stationarity in the next subsection).

It is worth mentioning that $\mathcal{F}_d(\Delta, L)$ contains a rather broad class of functions used in real-world application problems. Typical examples with additional regularity properties include Hadamard semi-differentiable functions [76, 32, 85], Whitney-stratifiable functions [13, 30], $o$-minimally definable functions [27] and a class of semi-algebraic functions [5, 30]. Thus, our gradient-free methods can be applied for solving these problems with finite-time convergence guarantees.

## 2.2 Generalized gradients and Goldstein subdifferential

We start with the definition of generalized gradients [26] for nondifferentiable functions. This is perhaps the most standard extension of gradients to nonsmooth and nonconvex functions.

**Definition 2.2** *Given a point $\mathbf{x} \in \mathbb{R}^d$ and a direction $\mathbf{v} \in \mathbb{R}^d$, the generalized directional derivative of a nondifferentiable function $f$ is given by $Df(\mathbf{x}; \mathbf{v}) := \limsup_{\mathbf{y} \to \mathbf{x}, t \downarrow 0} \frac{f(\mathbf{y} + t\mathbf{v}) - f(\mathbf{y})}{t}$. Then, the generalized gradient of $f$ is defined as a set $\partial f(\mathbf{x}) := \{\mathbf{g} \in \mathbb{R}^d : \mathbf{g}^\top \mathbf{v} \leq Df(\mathbf{x}; \mathbf{v}), \forall \mathbf{v} \in \mathbb{R}^d\}$.*

Rademacher's theorem guarantees that any Lipschitz function is almost everywhere differentiable. This implies that the generalized gradients of Lipschitz functions have additional properties and we can define them in a relatively simple way. The following proposition summarizes these results; we refer to Clarke [26] for the proof details.

**Proposition 2.1** *Suppose that $f$ is $L$-Lipschitz for some $L > 0$, we have that $\partial f(\mathbf{x})$ is a nonempty, convex and compact set and $\|\mathbf{g}\| \leq L$ for all $\mathbf{g} \in \partial f(\mathbf{x})$. Further, $\partial f(\cdot)$ is an upper-semicontinuous set-valued map. Moreover, a generalization of mean-value theorem holds: for any $\mathbf{x}_1, \mathbf{x}_2 \in \mathbb{R}^d$, there exist $\lambda \in (0, 1)$ and $\mathbf{g} \in \partial f(\lambda \mathbf{x}_1 + (1 - \lambda)\mathbf{x}_2)$ such that $f(\mathbf{x}_1) - f(\mathbf{x}_2) = \mathbf{g}^\top(\mathbf{x}_1 - \mathbf{x}_2)$. Finally, there is a simple way to represent the generalized gradient $\partial f(\mathbf{x})$:*

$$\partial f(\mathbf{x}) := \mathrm{conv}\left\{\mathbf{g} \in \mathbb{R}^d : \mathbf{g} = \lim_{\mathbf{x}_k \to \mathbf{x}} \nabla f(\mathbf{x}_k)\right\},$$

*which is the convex hull of all limit points of $\nabla f(\mathbf{x}_k)$ over all sequences $\mathbf{x}_1, \mathbf{x}_2, \ldots$ of differentiable points of $f(\cdot)$ which converge to $\mathbf{x}$.*

Given this definition of generalized gradients, a *Clarke stationary point* of $f$ is a point $\mathbf{x}$ satisfying $\mathbf{0} \in \partial f(\mathbf{x})$. Then, it is natural to ask if an optimization algorithm can reach an $\epsilon$-stationary point with a finite-time convergence guarantee. Here a point $\mathbf{x} \in \mathbb{R}^d$ is an $\epsilon$-Clarke stationary point if

$$\min\{\|\mathbf{g}\| : \mathbf{g} \in \partial f(\mathbf{x})\} \leq \epsilon.$$

This question has been addressed by [85, Theorem 1], who showed that finding an $\epsilon$-Clarke stationary points in nonsmooth nonconvex optimization can not be achieved by any finite-time algorithm given a fixed tolerance $\epsilon \in [0, 1)$. One possible response is to consider a relaxation called a *near $\epsilon$-Clarke stationary point*. Consider a point which is $\delta$-close to an $\epsilon$-stationary point for some $\delta > 0$. A point $\mathbf{x} \in \mathbb{R}^d$ is near $\epsilon$-stationary if the following statement holds true:

$$\min\left\{\|\mathbf{g}\| : \mathbf{g} \in \cup_{\mathbf{y} \in \mathbb{B}_\delta(\mathbf{x})} \partial f(\mathbf{y})\right\} \leq \epsilon.$$

Unfortunately, however, [58, Theorem 1] demonstrated that it is impossible to obtain worst-case guarantees for finding a near $\epsilon$-Clarke stationary point of $f \in \mathcal{F}_d(\Delta, L)$ when $\epsilon, \delta > 0$ are smaller than some certain constants unless the number of oracle calls has an exponential dependence on the problem dimension $d \geq 1$. These negative results suggest a need for rethinking the definition of targeted stationary points. We propose to consider the refined notion of Goldstein subdifferential.

**Definition 2.3** *Given a point $\mathbf{x} \in \mathbb{R}^d$ and $\delta > 0$, the $\delta$-Goldstein subdifferential of a Lipschitz function $f$ at $\mathbf{x}$ is given by $\partial_\delta f(\mathbf{x}) := \mathrm{conv}(\cup_{\mathbf{y} \in \mathbb{B}_\delta(\mathbf{x})} \partial f(\mathbf{y}))$.*

The Goldstein subdifferential of $f$ at $\mathbf{x}$ is the convex hull of the union of all generalized gradients at points in a $\delta$-ball around $\mathbf{x}$. Accordingly, we can define the $(\delta, \epsilon)$-Goldstein stationary points; that is, a point $\mathbf{x} \in \mathbb{R}^d$ is a $(\delta, \epsilon)$-Goldstein stationary point if the following statement holds:

$$\min\{\|\mathbf{g}\| : \mathbf{g} \in \partial_\delta f(\mathbf{x})\} \leq \epsilon.$$

It is worth mentioning that $(\delta, \epsilon)$-Goldstein stationarity is a weaker notion than (near) $\epsilon$-Clarke stationarity since any (near) $\epsilon$-stationary point is a $(\delta, \epsilon)$-Goldstein stationary point but not vice versa. However, the converse holds true under a smoothness condition [85, Proposition 6] and $\lim_{\delta \downarrow 0} \partial_\delta f(\mathbf{x}) = \partial f(\mathbf{x})$ holds as shown in Zhang et al. [85, Lemma 7]. The latter result also enables an intuitive framework for transforming nonasymptotic analysis of convergence to $(\delta, \epsilon)$-Goldstein stationary points to classical asymptotic results for finding $\epsilon$-Clarke stationary points. Thus, we conclude that finding a $(\delta, \epsilon)$-Goldstein stationary point is a reasonable optimality condition for general nonsmooth nonconvex optimization.

**Remark 2.2** *Finding a $(\delta, \epsilon)$-Goldstein stationary point in nonsmooth nonconvex optimization has been formally shown to be computationally tractable in an oracle model [85, 31, 80]. Goldstein [46] discovered that one can decrease the function value of a Lipschitz $f$ by using the minimal-norm element of $\partial_\delta f(\mathbf{x})$ and this leads to a deterministic normalized subgradient method which finds a $(\delta, \epsilon)$-Goldstein stationary point within $O(\frac{\Delta}{\delta \epsilon})$ iterations. However, Goldstein's algorithm is only conceptual since it is computationally intractable to return an exact minimal-norm element of $\partial_\delta f(\mathbf{x})$. Recently, the randomized variants of Goldstein's algorithm have been proposed with a convergence guarantee of $O(\frac{\Delta L^2}{\delta \epsilon^3})$ [85, 31, 80]. However, it remains unknown if gradient-free methods find a $(\delta, \epsilon)$-Goldstein stationary point of a Lipschitz function $f$ within $O(\mathrm{poly}(d, L, \Delta, 1/\epsilon, 1/\delta))$ iterations in the absence of an exact function value oracle. Note that the dependence on the problem dimension $d \geq 1$ is necessary for gradient-free methods as mentioned before.*

## 2.3 Randomized smoothing

The randomized smoothing approaches are simple and work equally well for convex and nonconvex functions. Formally, given the $L$-Lipschitz function $f$ (possibly nonsmooth nonconvex) and a distribution $\mathbb{P}$, we define $f_\delta(\mathbf{x}) = \mathbb{E}_{\mathbf{u} \sim \mathbb{P}}[f(\mathbf{x} + \delta \mathbf{u})]$. In particular, letting $\mathbb{P}$ be a standard Gaussian distribution, the function $f_\delta$ is a $\delta L \sqrt{d}$-approximation of $f(\cdot)$ and the gradient $\nabla f_\delta$ is $\frac{L\sqrt{d}}{\delta}$-Lipschitz where $d \geq 1$ is the problem dimension; see Nesterov and Spokoiny [72, Theorem 1 and Lemma 2]. Letting $\mathbb{P}$ be an uniform distribution on an unit ball in $\ell_2$-norm, the resulting function $f_\delta$ is a $\delta L$-approximation of $f(\cdot)$ and $\nabla f_\delta$ is also $\frac{cL\sqrt{d}}{\delta}$-Lipschitz where $d \geq 1$ is the problem dimension; see Yousefian et al. [84, Lemma 8] and Duchi et al. [36, Lemma E.2], rephrased as follows.

**Proposition 2.3** *Let $f_\delta(\mathbf{x}) = \mathbb{E}_{\mathbf{u} \sim \mathbb{P}}[f(\mathbf{x} + \delta \mathbf{u})]$ where $\mathbb{P}$ is an uniform distribution on an unit ball in $\ell_2$-norm. Assuming that $f$ is $L$-Lipschitz, we have (i) $|f_\delta(\mathbf{x}) - f(\mathbf{x})| \leq \delta L$, and (ii) $f_\delta$ is differentiable and $L$-Lipschitz with the $\frac{cL\sqrt{d}}{\delta}$-Lipschitz gradient where $c > 0$ is a constant. In addition, there exists a function $f$ for which each of the above bounds are tight simultaneously.*

The randomized smoothing approaches form the basis for developing gradient-free methods [39, 1, 2, 44, 72]. Given an access to function values of $f$, we can compute an unbiased estimate of the gradient of $f_\delta$ and plug them into stochastic gradient-based methods. Note that the Lipschitz constant of $f_\delta$ depends on the problem dimension $d \geq 1$ with at least a factor of $\sqrt{d}$ for many randomized smoothing approaches [58, Theorem 2]. This is consistent with the lower bounds for all gradient-free methods in convex and strongly convex optimization [37, 75].

# 3 Main Results

We establish a relationship between the Goldstein subdifferential and the uniform smoothing approach. We propose a gradient-free method (GFM), its stochastic variant (SGFM), and a two-phase version of GFM and SGFM. We analyze these algorithms using the Goldstein subdifferential; we provide the global rate and large-deviation estimates in terms of $(\delta, \epsilon)$-Goldstein stationarity.

## 3.1 Linking Goldstein subdifferential to uniform smoothing

Recall that $\partial_\delta f$ and $f_\delta$ are defined by $\partial_\delta f(\mathbf{x}) := \mathrm{conv}(\cup_{\mathbf{y} \in \mathbb{B}_\delta(\mathbf{x})} \partial f(\mathbf{y}))$ and $f_\delta(\mathbf{x}) = \mathbb{E}_{\mathbf{u} \sim \mathbb{P}}[f(\mathbf{x} + \delta \mathbf{u})]$. It is clear that $f$ is almost everywhere differentiable since $f$ is $L$-Lipschitz. This implies that $\nabla f_\delta(\mathbf{x}) = \mathbb{E}_{\mathbf{u} \sim \mathbb{P}}[\nabla f(\mathbf{x} + \delta \mathbf{u})]$ and demonstrates that $\nabla f_\delta(\mathbf{x})$ can be viewed intuitively as a convex combination of $\nabla f(\mathbf{z})$ over an infinite number of points $\mathbf{z} \in \mathbb{B}_\delta(\mathbf{x})$. As such, it is reasonable to conjecture that $\nabla f_\delta(\mathbf{x}) \in \partial_\delta f(\mathbf{x})$ for any $\mathbf{x} \in \mathbb{R}^d$. However, the above argument is not a rigorous proof; indeed, we need to justify why $\nabla f_\delta(\mathbf{x}) = \mathbb{E}_{\mathbf{u} \sim \mathbb{P}}[\nabla f(\mathbf{x} + \delta \mathbf{u})]$ if $f$ is almost everywhere differentiable and generalize the idea of a convex combination to include infinite sums. To resolve these issues, we exploit a toolbox due to Rockafellar and Wets [74].

In the following theorem, we summarize our result and refer to Appendix C for the proof details.

**Theorem 3.1** *Suppose that $f$ is $L$-Lipschitz and let $f_\delta(\mathbf{x}) = \mathbb{E}_{\mathbf{u} \sim \mathbb{P}}[f(\mathbf{x} + \delta \mathbf{u})]$, where $\mathbb{P}$ is an uniform distribution on a unit ball in $\ell_2$-norm and let $\partial_\delta f$ be a $\delta$-Goldstein subdifferential of $f$ (cf. Definition 2.3). Then, we have $\nabla f_\delta(\mathbf{x}) \in \partial_\delta f(\mathbf{x})$ for any $\mathbf{x} \in \mathbb{R}^d$.*

Theorem 3.1 resolves an important question and forms the basis for analyzing our gradient-free methods. Notably, our analysis can be extended to justify other randomized smoothing approaches in nonsmooth nonconvex optimization. For example, Nesterov and Spokoiny [72] used Gaussian smoothing and estimated the number of iterations required by their methods to output $\hat{\mathbf{x}} \in \mathbb{R}^d$ satisfying $\|\nabla f_\delta(\hat{\mathbf{x}})\| \leq \epsilon$. By modifying the proof of Theorem 3.1 and Zhang et al. [85, Lemma 7], we can prove that $\nabla f_\delta$ belongs to Goldstein subdifferential with Gaussian weights and this subdifferential converges to the Clarke subdifferential as $\delta \to 0$. Compared to uniform smoothing and the original Goldstein subdifferential, the proof for Gaussian smoothing is quite long and technical [72, Page 554], and adding Gaussian weights seems unnatural in general.

---

**Algorithm 1** Gradient-Free Method (GFM)

---

1: **Input:** initial point $\mathbf{x}^0 \in \mathbb{R}^d$, stepsize $\eta > 0$, problem dimension $d \geq 1$, smoothing parameter $\delta$ and iteration number $T \geq 1$.
2: **for** $t = 0, 1, 2, \ldots, T - 1$ **do**
3:     Sample $\mathbf{w}^t \in \mathbb{R}^d$ uniformly from a unit sphere in $\mathbb{R}^d$.
4:     Compute $\mathbf{g}^t = \frac{d}{2\delta}(f(\mathbf{x}^t + \delta\mathbf{w}^t) - f(\mathbf{x}^t - \delta\mathbf{w}^t))\mathbf{w}^t$.
5:     Compute $\mathbf{x}^{t+1} = \mathbf{x}^t - \eta\mathbf{g}^t$.
6: **Output:** $\mathbf{x}^R$ where $R \in \{0, 1, 2, \ldots, T - 1\}$ is uniformly sampled.

---

**Algorithm 2** Two-Phase Gradient-Free Method (2-GFM)

---

1: **Input:** initial point $\mathbf{x}^0 \in \mathbb{R}^d$, stepsize $\eta > 0$, problem dimension $d \geq 1$, smoothing parameter $\delta$, iteration number $T \geq 1$, number of rounds $S \geq 1$ and sample size $B$.
2: **for** $s = 0, 1, 2, \ldots, S - 1$ **do**
3:     Call Algorithm 1 with $\mathbf{x}^0$, $\eta$, $d$, $\delta$ and $T$ and let $\bar{\mathbf{x}}_s$ be an output.
4: **for** $s = 0, 1, 2, \ldots, S - 1$ **do**
5:     **for** $k = 0, 1, 2, \ldots, B - 1$ **do**
6:         Sample $\mathbf{w}^k \in \mathbb{R}^d$ uniformly from a unit sphere in $\mathbb{R}^d$.
7:         Compute $\mathbf{g}_s^k = \frac{d}{2\delta}(f(\bar{\mathbf{x}}_s + \delta\mathbf{w}^k) - f(\bar{\mathbf{x}}_s - \delta\mathbf{w}^k))\mathbf{w}^k$.
8:     Compute $\mathbf{g}_s = \frac{1}{B}\sum_{k=0}^{B-1} \mathbf{g}_s^k$.
9: Choose an index $s^\star \in \{0, 1, 2, \ldots, S - 1\}$ such that $s^\star = \operatorname{argmin}_{s=0,1,2,\ldots,S-1} \|\mathbf{g}_s\|$.
10: **Output:** $\bar{\mathbf{x}}_{s^\star}$.

---

## 3.2 Gradient-free methods

We analyze a gradient-free method (GFM) and its two-phase version (2-GFM) for optimizing a Lipschitz function $f$. Due to space limitations, we defer the proof details to Appendix D.

**Global rate estimation.** Let $f : \mathbb{R}^d \mapsto \mathbb{R}$ be a $L$-Lipschitz function and the smooth version of $f$ is then the function $f_\delta = \mathbb{E}_{\mathbf{u}\sim\mathbb{P}}[f(\mathbf{x} + \delta\mathbf{u})]$ where $\mathbb{P}$ is an uniform distribution on an unit ball in $\ell_2$-norm. Equipped with Lemma 10 from Shamir [75], we can compute an unbiased estimator for the gradient $\nabla f_\delta(\mathbf{x}^t)$ using function values.

This leads to the gradient-free method (GFM) in Algorithm 1 that simply performs a one-step gradient descent to obtain $\mathbf{x}^t$. It is worth mentioning that we use a random iteration count $R$ to terminate the execution of Algorithm 1 and this will guarantee that GFM is valid. Indeed, we only derive that $\min_{t=1,2,\ldots,T} \|\nabla f_\delta(\mathbf{x}^t)\| \leq \epsilon$ in the theoretical analysis (see also Nesterov and Spokoiny [72, Section 7]) and finding the best solution from $\{\mathbf{x}^1, \mathbf{x}^2, \ldots, \mathbf{x}^T\}$ is difficult since the quantity $\|\nabla f_\delta(\mathbf{x}^t)\|$ is unknown. To estimate them using Monte Carlo simulation would incur additional approximation errors and raise some reliability issues. The idea of random output is not new but has been used by Ghadimi and Lan [44] for smooth and nonconvex stochastic optimization. Such scheme also gives us a computational gain with a factor of two in expectation.

**Theorem 3.2** *Suppose that $f$ is $L$-Lipschitz and let $\delta > 0$ and $0 < \epsilon < 1$. Then, there exists some $T > 0$ such that the output of Algorithm 1 with $\eta = \frac{1}{10}\sqrt{\frac{\delta(\Delta + \delta L)}{cd^{3/2}L^3T}}$ satisfies that $\mathbb{E}[\min\{\|\mathbf{g}\| : \mathbf{g} \in \partial_\delta f(\mathbf{x}^R)\}] \leq \epsilon$ and the total number of calls of the function value oracle is bounded by*

$$O\left(d^{\frac{3}{2}}\left(\frac{L^4}{\epsilon^4} + \frac{\Delta L^3}{\delta\epsilon^4}\right)\right),$$

*where $d \geq 1$ is the problem dimension, $L > 0$ is the Lipschtiz parameter of $f$ and $\Delta > 0$ is an upper bound for the initial objective function gap, $f(\mathbf{x}^0) - \inf_{\mathbf{x}\in\mathbb{R}^d} f(\mathbf{x}) > 0$.*

**Remark 3.3** *Theorem 3.2 illustrates the difference between gradient-based and gradient-free methods in nonsmooth nonconvex optimization. Indeed, Davis et al. [31] has recently proved the rate of $\tilde{O}(\delta^{-1}\epsilon^{-3})$ for a randomized gradient-based method in terms of $(\delta, \epsilon)$-Goldstein stationarity. Further, Theorem 3.2 demonstrates that nonsmooth nonconvex optimization is likely to be intrinsically harder than all other standard settings. More specifically, the state-of-the-art rate for gradient-free methods is $O(d\epsilon^{-2})$ for nonsmooth convex optimization in terms of objective function value gap [37] and smooth nonconvex optimization in terms of gradient norm [72]. Thus, the dependence on $d \geq 1$ is*

*linear in their bounds yet $d^{\frac{3}{2}}$ in our bound. We believe it is promising to either improve the rate of gradient-free methods or show the impossibility by establishing a lower bound.*

**Large-deviation estimation.** While Theorem 3.2 establishes the expected convergence rate over many runs of Algorithm 1, we are also interested in the large-deviation properties for a single run. Indeed, we hope to establish a complexity bound for computing a $(\delta, \epsilon, \Lambda)$-*solution*; that is, a point $\mathbf{x} \in \mathbb{R}^d$ satisfying $\text{Prob}(\min\{\|\mathbf{g}\| : \mathbf{g} \in \partial_\delta f(\mathbf{x})\} \leq \epsilon) \geq 1 - \Lambda$ for some $\delta > 0$ and $0 < \epsilon, \Lambda < 1$. By Theorem 3.2 and Markov's inequality,

$$\text{Prob}\left(\min\{\|\mathbf{g}\| : \mathbf{g} \in \partial_\delta f(\mathbf{x}^R)\} \geq \lambda \mathbb{E}[\min\{\|\mathbf{g}\| : \mathbf{g} \in \partial_\delta f(\mathbf{x}^R)\}]\right) \leq \frac{1}{\lambda}, \quad \text{for all } \lambda > 0,$$

we conclude that the total number of calls of the function value oracle is bounded by

$$O\left(d^{\frac{3}{2}}\left(\frac{L^4}{\Lambda^4 \epsilon^4} + \frac{\Delta L^3}{\delta \Lambda^4 \epsilon^4}\right)\right). \tag{3.1}$$

This complexity bound is rather pessimistic in terms of its dependence on $\Lambda$ which is often set to be small in practice. To improve the bound, we combine Algorithm 1 with a post-optimization procedure [44], leading to a two-phase gradient-free method (2-GFM), shown in Algorithm 2.

**Theorem 3.4** *Suppose that $f$ is $L$-Lipschitz and let $\delta > 0$ and $0 < \epsilon, \Lambda < 1$. Then, there exists some $T, S, B > 0$ such that the output of Algorithm 2 with $\eta = \frac{1}{10}\sqrt{\frac{\delta(\Delta + \delta L)}{cd^{3/2}L^3T}}$ satisfies that $\text{Prob}(\min\{\|\mathbf{g}\| : \mathbf{g} \in \partial_\delta f(\bar{\mathbf{x}}_{s^\star})\}] \geq \epsilon) \leq \Lambda$ and the total number of calls of the function value oracle is bounded by*

$$O\left(d^{\frac{3}{2}}\left(\frac{L^4}{\epsilon^4} + \frac{\Delta L^3}{\delta \epsilon^4}\right)\log_2\left(\frac{1}{\Lambda}\right) + \frac{dL^2}{\Lambda \epsilon^2}\log_2\left(\frac{1}{\Lambda}\right)\right),$$

*where $d \geq 1$ is the problem dimension, $L > 0$ is the Lipschtiz parameter of $f$ and $\Delta > 0$ is an upper bound for the initial objective function gap, $f(\mathbf{x}^0) - \inf_{\mathbf{x} \in \mathbb{R}^d} f(\mathbf{x}) > 0$.*

Clearly, the bound in Theorem 3.4 is significantly smaller than the corresponding one in Eq. (3.1) in terms of the dependence on $1/\Lambda$, demonstrating the power of the post-optimization phase.

### 3.3 Stochastic gradient-free methods

We turn to the analysis of a stochastic gradient-free method (SGFM) and its two-phase version (2-SGFM) for optimizing a Lipschitz function $f(\cdot) = \mathbb{E}_{\xi \in \mathbb{P}_\mu}[F(\cdot, \xi)]$.

**Global rate estimation.** In contrast to minimizing a deterministic function $f$, we only have access to the noisy function value $F(\mathbf{x}, \xi)$ at any point $\mathbf{x} \in \mathbb{R}^d$ where a data sample $\xi$ is drawn from a distribution $\mathbb{P}_\mu$. Intuitively, this is a more challenging setup. It has been studied before in the setting of optimizing a nonsmooth convex function [37, 72] or a smooth nonconvex function [44]. As in these papers, we assume that (i) $F(\cdot, \xi)$ is $L(\xi)$-Lipschitz with $\mathbb{E}_{\xi \in \mathbb{P}_\mu}[L^2(\xi)] \leq G^2$ for some $G > 0$ and (ii) $\mathbb{E}[F(\mathbf{x}, \xi^t)] = f(\mathbf{x})$ for all $\mathbf{x} \in \mathbb{R}^d$ where $\xi^t$ is simulated from $\mathbb{P}_\mu$ at the $t^{\text{th}}$ iteration.

Despite the noisy function value, we can compute an unbiased estimator of the gradient $\nabla f_\delta(\mathbf{x}^t)$, where $f_\delta = \mathbb{E}_{\mathbf{u} \sim \mathbb{P}}[f(\mathbf{x} + \delta \mathbf{u})] = \mathbb{E}_{\mathbf{u} \sim \mathbb{P}, \xi \in \mathbb{P}_\mu}[F(\mathbf{x} + \delta \mathbf{u}, \xi)]$. In particular, we have $\hat{\mathbf{g}}^t = \frac{d}{2\delta}(F(\mathbf{x}^t + \delta \mathbf{w}^t, \xi^t) - F(\mathbf{x}^t - \delta \mathbf{w}^t, \xi^t))\mathbf{w}^t$. Clearly, under our assumption, we have

$$\mathbb{E}_{\mathbf{u} \sim \mathbb{P}, \xi \in \mathbb{P}_\mu}[\hat{\mathbf{g}}^t] = \mathbb{E}_{\mathbf{u} \sim \mathbb{P}}[\mathbb{E}_{\xi \in \mathbb{P}_\mu}[\hat{\mathbf{g}}^t \mid \mathbf{u}]] = \mathbb{E}_{\mathbf{u} \sim \mathbb{P}}[\mathbf{g}^t] = \nabla f_\delta(\mathbf{x}^t),$$

where $\mathbf{g}^t$ is defined in Algorithm 1. However, the variance of the estimator $\hat{\mathbf{g}}^t$ can be undesirably large since $F(\cdot, \xi)$ is $L(\xi)$-Lipschitz for a (possibly unbounded) random variable $L(\xi) > 0$. To resolve this issue, we revisit Shamir [75, Lemma 10] and show that in deriving an upper bound for $\mathbb{E}_{\mathbf{u} \sim \mathbb{P}, \xi \in \mathbb{P}_\mu}[\|\hat{\mathbf{g}}^t\|^2]$ it suffices to assume that $\mathbb{E}_{\xi \in \mathbb{P}_\mu}[L^2(\xi)] \leq G^2$ for some constant $G > 0$. The resulting bound achieves a linear dependence in the problem dimension $d > 0$ which is the same as in Shamir [75, Lemma 10]. Note that the setup with *convex* and $L(\xi)$-Lipschitz functions $F(\cdot, \xi)$ has been considered in Duchi et al. [37]. However, our estimator is different from their estimator of $\hat{\mathbf{g}}^t = \frac{d}{\delta}(F(\mathbf{x}^t + \delta \mathbf{w}^t, \xi^t) - F(\mathbf{x}^t, \xi^t))\mathbf{w}^t$ which essentially suffers from the quadratic dependence in $d > 0$. It is also necessary to employ a random iteration count $R$ to terminate Algorithm 3.

---

**Algorithm 3** Stochastic Gradient-Free Method (SGFM)

---

1: **Input:** initial point $\mathbf{x}^0 \in \mathbb{R}^d$, stepsize $\eta > 0$, problem dimension $d \geq 1$, smoothing parameter $\delta$ and iteration number $T \geq 1$.
2: **for** $t = 0, 1, 2, \ldots, T$ **do**
3:     Simulate $\xi^t$ from the distribution $\mathbb{P}_\mu$.
4:     Sample $\mathbf{w}^t \in \mathbb{R}^d$ uniformly from a unit sphere in $\mathbb{R}^d$.
5:     Compute $\hat{\mathbf{g}}^t = \frac{d}{2\delta}(F(\mathbf{x}^t + \delta \mathbf{w}^t, \xi^t) - F(\mathbf{x}^t - \delta \mathbf{w}^t, \xi^t))\mathbf{w}^t$.
6:     Compute $\mathbf{x}^{t+1} = \mathbf{x}^t - \eta \mathbf{g}^t$.
7: **Output:** $\mathbf{x}^R$ where $R \in \{0, 1, 2, \ldots, T-1\}$ is uniformly sampled.

---

---

**Algorithm 4** Two-Phase Stochastic Gradient-Free Method (2-SGFM)

---

1: **Input:** initial point $\mathbf{x}^0 \in \mathbb{R}^d$, stepsize $\eta > 0$, problem dimension $d \geq 1$, smoothing parameter $\delta$, iteration number $T \geq 1$, number of rounds $S \geq 1$ and sample size $B$.
2: **for** $s = 0, 1, 2, \ldots, S-1$ **do**
3:     Call Algorithm 3 with $\mathbf{x}^0$, $\eta$, $d$, $\delta$ and $T$ and let $\bar{\mathbf{x}}_s$ be an output.
4: **for** $s = 0, 1, 2, \ldots, S-1$ **do**
5:     **for** $k = 0, 1, 2, \ldots, B-1$ **do**
6:        Simulate $\xi^k$ from the distribution $\mathbb{P}_\mu$.
7:        Sample $\mathbf{w}^k \in \mathbb{R}^d$ uniformly from a unit sphere in $\mathbb{R}^d$.
8:        Compute $\hat{\mathbf{g}}_s^k = \frac{d}{2\delta}(F(\bar{\mathbf{x}}_s + \delta \mathbf{w}^k, \delta^k) - F(\bar{\mathbf{x}}_s - \delta \mathbf{w}^k, \delta^k))\mathbf{w}^k$.
9:     Compute $\hat{\mathbf{g}}_s = \frac{1}{B}\sum_{k=0}^{B-1} \hat{\mathbf{g}}_s^k$.
10: Choose an index $s^\star \in \{0, 1, 2, \ldots, S-1\}$ such that $s^\star = \operatorname{argmin}_{s=0,1,2,\ldots,S-1} \|\hat{\mathbf{g}}_s\|$.
11: **Output:** $\bar{\mathbf{x}}_{s^\star}$.

---

**Theorem 3.5** *Suppose that $F(\cdot, \xi)$ is $L(\xi)$-Lipschitz with $\mathbb{E}_{\xi \in \mathbb{P}_\mu}[L^2(\xi)] \leq G^2$ for some $G > 0$ and let $\delta > 0$ and $0 < \epsilon < 1$. Then, there exists some $T > 0$ such that the output of Algorithm 3 with $\eta = \frac{1}{10}\sqrt{\frac{\delta(\Delta+\delta G)}{cd^{3/2}G^3 T}}$ satisfies that $\mathbb{E}[\min\{\|\mathbf{g}\| : \mathbf{g} \in \partial_\delta f(\mathbf{x}^R)\}] \leq \epsilon$ and the total number of calls of the noisy function value oracle is bounded by*

$$O\left(d^{\frac{3}{2}}\left(\frac{G^4}{\epsilon^4} + \frac{\Delta G^3}{\delta \epsilon^4}\right)\right),$$

*where $d \geq 1$ is the problem dimension, $L > 0$ is the Lipschtiz parameter of $f$ and $\Delta > 0$ is an upper bound for the initial objective function gap, $f(\mathbf{x}^0) - \inf_{\mathbf{x} \in \mathbb{R}^d} f(\mathbf{x}) > 0$.*

In the stochastic setting, the gradient-based method achieves the rate of $O(\delta^{-1}\epsilon^{-4})$ for a randomized gradient-based method in terms of $(\delta, \epsilon)$-Goldstein stationarity [31]. As such, our bound in Theorem 3.5 is tight up to the problem dimension $d \geq 1$. Further, the state-of-the-art rate for stochastic gradient-free methods is $O(d\epsilon^{-2})$ for nonsmooth convex optimization in terms of objective function value gap [37] and $O(d\epsilon^{-4})$ for smooth nonconvex optimization in terms of gradient norm [44]. Thus, Theorem 3.5 demonstrates that nonsmooth nonconvex stochastic optimization is essentially the most difficult one among than all these standard settings.

**Large-deviation estimation.** As in the case of GFM, we hope to establish a complexity bound of SGFM for computing a $(\delta, \epsilon, \Lambda)$-solution. By Theorem 3.5 and Markov's inequality, we obtain that the total number of calls of the noisy function value oracle is bounded by

$$O\left(d^{\frac{3}{2}}\left(\frac{G^4}{\Lambda^4\epsilon^4} + \frac{\Delta G^3}{\delta \Lambda^4 \epsilon^4}\right)\right). \tag{3.2}$$

We also propose a two-phase stochastic gradient-free method (2-SGFM) in Algorithm 4 by combining Algorithm 3 with a post-optimization procedure.

**Theorem 3.6** *Suppose that $F(\cdot, \xi)$ is $L(\xi)$-Lipschitz with $\mathbb{E}_{\xi \in \mathbb{P}_\mu}[L^2(\xi)] \leq G^2$ for some $G > 0$ and let $\delta > 0$ and $0 < \epsilon, \Lambda < 1$. Then, there exists some $T, S, B > 0$ such that the output of Algorithm 4 with $\eta = \frac{1}{10}\sqrt{\frac{\delta(\Delta+\delta G)}{cd^{3/2}G^3 T}}$ satisfies that $\operatorname{Prob}(\min\{\|\mathbf{g}\| : \mathbf{g} \in \partial_\delta f(\bar{\mathbf{x}}_{s^\star})\} \geq \epsilon) \leq \Lambda$ and the total number of calls of the noisy function value oracle is bounded by*

$$O\left(d^{\frac{3}{2}}\left(\frac{G^4}{\epsilon^4} + \frac{\Delta G^3}{\delta \epsilon^4}\right)\log_2\left(\frac{1}{\Lambda}\right) + \frac{dG^2}{\Lambda\epsilon^2}\log_2\left(\frac{1}{\Lambda}\right)\right),$$

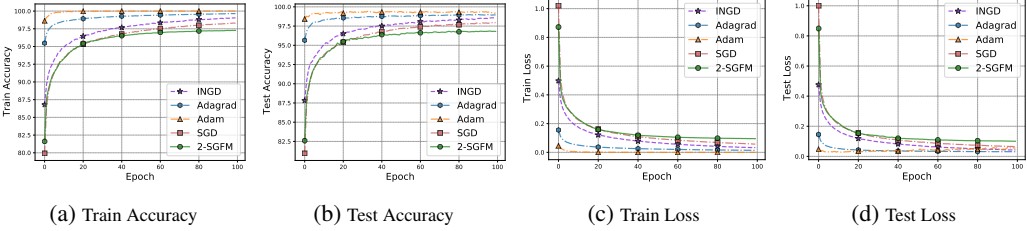

| (a) Train Accuracy | (b) Test Accuracy | (c) Train Loss | (d) Test Loss |

Figure 1: Performance of different methods on training CNNs with the MNIST dataset.

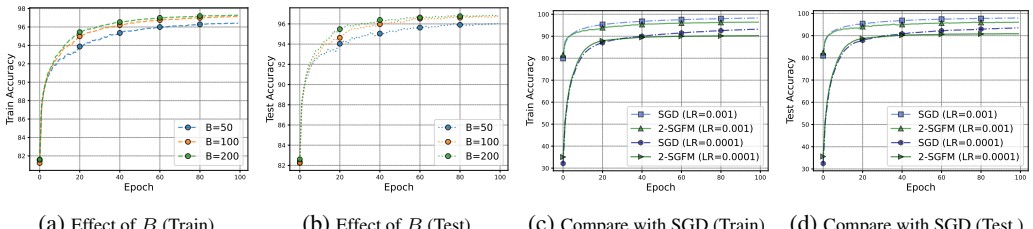

| (a) Effect of $B$ (Train) | (b) Effect of $B$ (Test) | (c) Compare with SGD (Train) | (d) Compare with SGD (Test) |

Figure 2: (**a-b**) Performance of 2-SGFM with different choices of $B$. (**c-d**) Performance of 2-SGFM and SGD with different choices of learning rates.

*where $d \geq 1$ is the problem dimension, $L > 0$ is the Lipschtiz parameter of $f$ and $\Delta > 0$ is an upper bound for the initial objective function gap $f(\mathbf{x}^0) - \inf_{\mathbf{x} \in \mathbb{R}^d} f(\mathbf{x}) > 0$.*

**Further discussions.** We remark that the choice of stepsize $\eta$ in all of our zeroth-order methods depend on $\Delta$, whereas such dependence is not necessary in the first-order setting; see e.g., Zhang et al. [85]. Setting the stepsize without any prior knowledge of $\Delta$, our methods can still achieve finite-time convergence guarantees but the order would become worse. This is possibly because the first-order information gives more characterization of the objective function than the zeroth-order information, so that for first-order methods the stepsize can be independent of more problem parameters without sacrificing the bound. A bit on the positive side is that, it suffices for our zeroth-order methods to know an estimate of the upper bound of $\Theta(\Delta)$, which can be done in certain application problems.

Moreover, we highlight that $\delta > 0$ is the desired tolerance in our setting. In fact, $(\delta, \epsilon)$-Goldstein stationarity (see Definition 2.3) relaxes $\epsilon$-Clarke stationarity and our methods pursue an $(\delta, \epsilon)$-stationary point since finding an $\epsilon$-Clarke point is intractable. This is different from smooth optimization where $\epsilon$-Clarke stationarity reduces to $\nabla f(\mathbf{x}) \leq \epsilon$ and becomes tractable. In this context, the existing zeroth-order methods are designed to pursue an $\epsilon$-stationary point. Notably, a $(\delta, \epsilon)$-Goldstein stationary point is provably an $\epsilon$-stationary point in smooth optimization if we choose $\delta$ that relies on $d$ and $\epsilon$.

## 4 Experiment

We conduct numerical experiments to validate the effectiveness of our proposed methods. In particular, we evaluate the performance of our two-phase version of SGFM (Algorithm 4) on the task of image classification using convolutional neural networks (CNNs) with ReLU activations. The dataset we use is the MNIST dataset[1] [60] and the CNN framework we use is: (i) we set two convolution layers and two fully connected layers where the dropout layers [77] are used before each fully connected layer, and (ii) two convolution layers and the first fully connected layer are associated with ReLU activation. It is worth mentioning that our setup follows the default one[2] and the similar setup was also consider in Zhang et al. [85] for evaluating the gradient-based methods (see the setups and results for CIFAR10 dataset in Appendix F).

The baseline approaches include three gradient-based methods: stochastic gradient descent (SGD), ADAGRAD [34] and ADAM [55]. We compare these methods with 2-SGFM (cf. Algorithm 4) and set the learning rate $\eta$ as 0.001. All the experiments are implemented using PyTorch [73] on a workstation with a 2.6 GHz Intel Core i7 and 16GB memory.

---

[1]http://yann.lecun.com/exdb/mnist

[2]https://github.com/pytorch/examples/tree/main/mnist

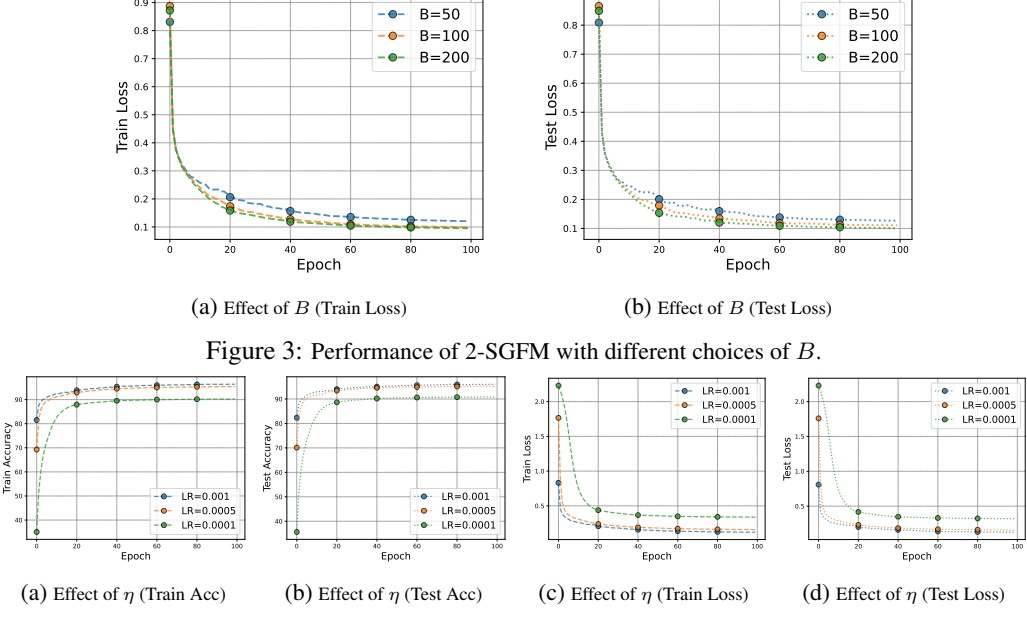

(a) Effect of $B$ (Train Loss)    (b) Effect of $B$ (Test Loss)

Figure 3: Performance of 2-SGFM with different choices of $B$.

(a) Effect of $\eta$ (Train Acc)    (b) Effect of $\eta$ (Test Acc)    (c) Effect of $\eta$ (Train Loss)    (d) Effect of $\eta$ (Test Loss)

Figure 4: Performance of 2-SGFM with different choices of learning rates.

Figure 1 summarizes the numerical results on the performance of SGD, ADAGRAD, Adagrad, ADAM, INDG [85], and our method 2-SGFM with $\delta = 0.1$ and $B = 200$. Notably, 2-SGFM is comparable to other gradient-based methods in terms of training/test accuracy/loss even though it only use the function values. This demonstrates the potential value of our methods since the gradient-based methods are not applicable in many real-world application problems as mentioned before. Figure 2a and 2b presents the effect of batch size $B \geq 1$ in 2-SGFM; indeed, the larger value of $B$ leads to better performance and this accords with Theorem 3.6. We also compare the performance of SGD and 2-SGFM with different choices of $\eta$. From Figure 2c and 2d, we see that SGD and 2-SGFM achieve similar performance in the early stage and converge to solutions with similar quality.

Figure 3 summarizes the experimental results on the effect of batch size $B$ for 2-SGFM. Note that the evaluation metrics here are train loss and test loss. It is clear that the larger value of $B$ leads to better performance and this is consistent with the results presented in the main context. Figure 4 summarizes the experimental results on the effect of learning rates for 2-SGFM. It is interesting to see that 2-SGFM can indeed benefit from a more aggressive choice of stepsize $\eta > 0$ in practice and the choice of $\eta = 0.0001$ seems to be too conservative.

## 5    Conclusion

We proposed and analyzed a class of deterministic and stochastic gradient-free methods for optimizing a Lipschitz function. Based on the relationship between the Goldstein subdifferential and uniform smoothing that we have established, the proposed GFM and SGFM are proved to return a $(\delta, \epsilon)$-Goldstein stationary point at an expected rate of $O(d^{3/2}\delta^{-1}\epsilon^{-4})$. We also obtain a large-deviation guarantee and improve it by combining GFM and SGFM with a two-phase scheme. Experiments on training neural networks with the MNIST and CIFAR10 datasets demonstrate the effectiveness of our methods. Future directions include the theory for non-Lipschitz and nonconvex optimization [11] and applications of our methods to deep residual neural network (ResNet) [47] and deep dense convolutional network (DenseNet) [50].

## Acknowledgements

We would like to thank the area chair and three anonymous referees for constructive suggestions that improve the paper. This work is supported in part by the Mathematical Data Science program of the Office of Naval Research under grant number N00014-18-1-2764 and by the Vannevar Bush Faculty Fellowship program under grant number N00014-21-1-2941.

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
