# A    Further Related Work on Nonsmooth Nonconvex Optimization

To appreciate the difficulty and the broad scope of the research agenda in nonsmooth nonconvex optimization, we start by describing the existing relevant literature. First, the existing work is mostly devoted to establishing the asymptotic convergence properties of various optimization algorithms, including gradient sampling (GS) methods [16–18, 57, 19], bundle methods [56, 40] and subgradient methods [8, 65, 30, 28, 12]. More specifically, Burke et al. [16] provided a systematic investigation of approximating the Clarke subdifferential through random sampling and proposed a gradient bundle method [17]—the precursor of GS methods—for optimizing a nonconvex, nonsmooth and non-Lipschitz function. Later, Burke et al. [18] and Kiwiel [57] proposed the GS methods by incorporating key modifications into the algorithmic scheme in Burke et al. [17] and proved that every cluster point of the iterates generated by GS methods is a Clarke stationary point. For an overview of GS methods, we refer to Burke et al. [19]. Another line of works extended the bundle methods to nonsmooth nonconvex optimization by considering either piece-wise linear models embedding possible downward shifting [56] or a mixture of linear pieces that exhibit a convex or concave behavior [40]. There has been recent progress on analyzing subgradient methods for nonsmooth nonconvex optimization; indeed, the classical subgradient method on Lipschitz functions may fail to asymptotically find any stationary point due to the pathological examples [28]. Under some additional regularity conditions, Benaïm et al. [8] proved the asymptotic convergence of stochastic approximation methods from a continuous-time viewpoint and Majewski et al. [65] generalized these results with proximal and implicit updates. Bolte and Pauwels [12] justify the automatic differentiation schemes under the nonsmoothness conditions; Davis et al. [30] proved the asymptotic convergence of classical subgradient methods for a class of Whitney stratifiable functions which include the functions studied in Majewski et al. [65]. Recently, Zhang et al. [85] modified Goldstein's subgradient method [46] to optimize a class of Hadamard directionally differentiable function and proved nonasymptotic convergence guarantee. Davis et al. [31] relaxed the assumption of Hadamard directionally differentiability and showed that another modification of Goldstein's subgradient method could achieve the same finite-time guarantee for any Lipschitz function. Concurrently, Tian et al. [80] removed the subgradient selection oracle assumption in Zhang et al. [85, Assumption 1] and provided the third modification of Goldstein's subgradient method with the same finite-time convergence. Different from these gradient-based methods, we focus on the gradient-free methods in this paper.

We are also aware of many recent works on the algorithmic design in the structured nonsmooth nonconvex optimization. There are two primary settings where the proximal gradient methods is guaranteed to achieve nonasymptotic convergence if the proximal mapping can be efficiently evaluated. The former one considers the objective function with composition structure [35, 33, 29], while the latter one focuses on composite objective functions with nonsmooth convex component [14, 7]. However, both of these settings require the weak convexity of objective function and exclude many simple and important nonsmooth nonconvex functions used in the real-world application problems.

# B    Proof of Proposition 2.3

Throughout this subsection, we let $\mathbf{u} \in \mathbb{R}^d$ denote a random variable distributed uniformly on $\mathbb{B}_1(\mathbf{0})$. For the first statement, since $f$ is $L$-Lipschitz, we have
$$|f_\delta(\mathbf{x}) - f(\mathbf{x})| = |\mathbb{E}[f(\mathbf{x} + \delta\mathbf{u}) - f(\mathbf{x})]| \leq \delta L \cdot \mathbb{E}[\|\mathbf{u}\|] \leq \delta L.$$
Then, we proceed to prove the second statement. Indeed, Bertsekas [9, Proposition 2.4] guarantees that $f_\delta$ is everywhere differentiable. Since $f$ is $L$-Lipschitz, we have
$$|f_\delta(\mathbf{x}) - f_\delta(\mathbf{x}')| = |\mathbb{E}[f(\mathbf{x} + \delta\mathbf{u}) - f(\mathbf{x}' + \delta\mathbf{u})]| \leq L\mathbb{E}[\|\mathbf{x} - \mathbf{x}'\|] = L\|\mathbf{x} - \mathbf{x}'\|, \quad \text{for all } \mathbf{x}, \mathbf{x}' \in \mathbb{R}^d.$$
It remains to prove that $\nabla f_\delta$ is Lipschitz. Since $f$ is $L$-Lipschitz, the Rademacher's theorem guarantees that $f$ is almost everywhere differentiable. This implies that $\nabla f_\delta(\mathbf{x}) = \mathbb{E}[\nabla f(\mathbf{x} + \delta\mathbf{u})]$. Then, we have
$$\|\nabla f_\delta(\mathbf{x}) - \nabla f_\delta(\mathbf{x}')\| = \|\mathbb{E}[\nabla f(\mathbf{x} + \delta\mathbf{u})] - \mathbb{E}[\nabla f(\mathbf{x}' + \delta\mathbf{u})]\|$$
$$= \frac{1}{\text{Vol}(\mathbb{B}_1(\mathbf{0}))} \left| \int_{\mathbf{u} \in \mathbb{B}_1(\mathbf{0})} \nabla f(\mathbf{x} + \delta\mathbf{u}) \, d\mathbf{u} - \int_{\mathbf{u} \in \mathbb{B}_1(\mathbf{0})} \nabla f(\mathbf{x}' + \delta\mathbf{u}) \, d\mathbf{u} \right|$$
$$= \frac{1}{\text{Vol}(\mathbb{B}_\delta(\mathbf{0}))} \left| \int_{\mathbf{y} \in \mathbb{B}_\delta(\mathbf{x})} \nabla f(\mathbf{y}) \, d\mathbf{y} - \int_{\mathbf{y} \in \mathbb{B}_\delta(\mathbf{x}')} \nabla f(\mathbf{y}) \, d\mathbf{y} \right|.$$

Note that $f$ is $L$-Lipschitz, we have $\|\nabla f(\mathbf{y})\| \leq L$ for any $\mathbf{y} \in \mathbb{B}_\delta(\mathbf{x}) \cup \mathbb{B}_\delta(\mathbf{x}')$. Then, we turn to prove that $\|\nabla f_\delta(\mathbf{x}) - \nabla f_\delta(\mathbf{x}')\| \leq \frac{L\sqrt{d}\|\mathbf{x}-\mathbf{x}'\|}{\delta}$ for two different cases one by one as follows.

**Case I: $\|\mathbf{x} - \mathbf{x}'\| \geq 2\delta$.** It is clear that

$$\|\nabla f_\delta(\mathbf{x}) - \nabla f_\delta(\mathbf{x}')\| \leq 2L \leq \frac{L\|\mathbf{x}-\mathbf{x}'\|}{\delta} \stackrel{d \geq 1}{\leq} \frac{L\sqrt{d}\|\mathbf{x}-\mathbf{x}'\|}{\delta},$$

which implies the desired result.

**Case II: $\|\mathbf{x} - \mathbf{x}'\| \leq 2\delta$.** It is clear that $\mathbb{B}_\delta(\mathbf{x}) \cap \mathbb{B}_\delta(\mathbf{x}') \neq \emptyset$. This implies that

$$\|\nabla f_\delta(\mathbf{x}) - \nabla f_\delta(\mathbf{x}')\| = \frac{1}{\mathrm{Vol}(\mathbb{B}_\delta(\mathbf{0}))} \left| \int_{\mathbf{y} \in \mathbb{B}_\delta(\mathbf{x}) \setminus \mathbb{B}_\delta(\mathbf{x}')} \nabla f(\mathbf{y}) \, d\mathbf{y} - \int_{\mathbf{y} \in \mathbb{B}_\delta(\mathbf{x}') \setminus \mathbb{B}_\delta(\mathbf{x})} \nabla f(\mathbf{y}) \, d\mathbf{y} \right|.$$

Since $\|\nabla f(\mathbf{y})\| \leq L$ for any $\mathbf{y} \in \mathbb{B}_\delta(\mathbf{x}) \cup \mathbb{B}_\delta(\mathbf{x}')$, we have

$$\|\nabla f_\delta(\mathbf{x}) - \nabla f_\delta(\mathbf{x}')\| \leq \frac{L}{\mathrm{Vol}(\mathbb{B}_\delta(\mathbf{0}))} \left( \mathrm{Vol}(\mathbb{B}_\delta(\mathbf{x}) \setminus \mathbb{B}_\delta(\mathbf{x}')) + \mathrm{Vol}(\mathbb{B}_\delta(\mathbf{x}') \setminus \mathbb{B}_\delta(\mathbf{x})) \right).$$

By the symmetry from a geometrical point of view, we have $\mathrm{Vol}(\mathbb{B}_\delta(\mathbf{x}) \setminus \mathbb{B}_\delta(\mathbf{x}')) = \mathrm{Vol}(\mathbb{B}_\delta(\mathbf{x}') \setminus \mathbb{B}_\delta(\mathbf{x}))$. For simplicity, we let $I = \mathbb{B}_\delta(\mathbf{x}) \setminus \mathbb{B}_\delta(\mathbf{x}')$ and obtain that

$$\|\nabla f_\delta(\mathbf{x}) - \nabla f_\delta(\mathbf{x}')\| \leq \frac{2L}{\mathrm{Vol}(\mathbb{B}_\delta(\mathbf{0}))} \mathrm{Vol}(I) = \frac{2L}{c_d \delta^d} \mathrm{Vol}(I), \quad \text{where } c_d = \frac{\pi^{d/2}}{\Gamma(d/2+1)}.$$

It suffices to find an upper bound for $\mathrm{Vol}(I)$ in terms of $\|\mathbf{x} - \mathbf{x}'\|$. Let $V_{cap}(p)$ denote the volume of the spherical cap with the distance $p$ from the center of the sphere, we have

$$\mathrm{Vol}(I) = \mathrm{Vol}(\mathbb{B}_\delta(\mathbf{0})) - 2V_{cap}(\tfrac{1}{2}\|\mathbf{x}-\mathbf{x}'\|) = c_d \delta^d - 2V_{cap}(\tfrac{1}{2}\|\mathbf{x}-\mathbf{x}'\|).$$

The volume of the $d$-dimensional spherical cap with distance $p$ from the center of the sphere can be calculated in terms of the volumes of $(d-1)$-dimensional spheres as follows:

$$V_{cap}(p) = \int_p^\delta c_{d-1}(\delta^2 - \rho^2)^{\frac{d-1}{2}} \, d\rho, \quad \text{for all } p \in [0, \delta].$$

Since $V_{cap}(\cdot)$ is a convex function over $[0, \delta]$, we have $V_{cap}(p) \geq V_{cap}(0) + V'_{cap}(0)p$. By the definition, we have $V_{cap}(0) = \frac{1}{2}\mathrm{Vol}(\mathbb{B}_\delta(\mathbf{0})) = \frac{1}{2}c_d\delta^d$ and $V'_{cap}(0) = -c_{d-1}\delta^{d-1}$. Thus, $V_{cap}(p) \geq \frac{1}{2}c_d\delta^d - c_{d-1}\delta^{d-1}p$. Furthermore, $\frac{1}{2}\|\mathbf{x}-\mathbf{x}'\| \in [0, \delta]$. Putting these pieces together yields that $\mathrm{Vol}(I) \leq c_{d-1}\delta^{d-1}\|\mathbf{x} - \mathbf{x}'\|$. Therefore, we conclude that

$$\|\nabla f_\delta(\mathbf{x}) - \nabla f_\delta(\mathbf{x}')\| \leq \frac{2L}{c_d\delta^d} \mathrm{Vol}(I) \leq \frac{2c_{d-1}}{c_d} \frac{L\|\mathbf{x}-\mathbf{x}'\|}{\delta}.$$

Since $c_d = \frac{\pi^{d/2}}{\Gamma(d/2+1)}$, we have $\frac{2c_{d-1}}{c_d} = \begin{cases} \frac{d!!}{(d-1)!!} & \text{if } d \text{ is odd,} \\ \frac{2}{\pi} \frac{d!!}{(d-1)!!} & \text{otherwise.} \end{cases}$ and $\frac{1}{\sqrt{d}} \frac{2c_{d-1}}{c_d} \to \sqrt{\frac{\pi}{2}}$. Therefore,

we conclude that the gradient $\nabla f_\delta$ is $\frac{cL\sqrt{d}}{\delta}$-Lipschitz where $c > 0$ is a positive constant. In addition, for the construction of a function $f$ in which each of the above bounds are tight, we consider a convex combination of "difficult" functions, in this case

$$f_1(\mathbf{x}) = L\|\mathbf{x}\|, \quad f_2(\mathbf{x}) = L|\langle \mathbf{x}, \tfrac{\mathbf{w}}{\|\mathbf{w}\|} \rangle - \tfrac{1}{2}|.$$

and choose $f(\mathbf{x}) = \frac{1}{2}(f_1(\mathbf{x}) + f_2(\mathbf{x}))$. Following up the same argument as in Duchi et al. [36, Lemma 10], it is relatively straightforward to verify that the bounds in Proposition 2.3 cannot be improved by more than a constant factor. This completes the proof.

## C   Proof of Theorem 3.1

We first show that $\nabla f_\delta(\mathbf{x}) = \mathbb{E}_{\mathbf{u} \sim \mathbb{P}}[\nabla f(\mathbf{x} + \delta\mathbf{u})]$. Indeed, by the definition of $f_\delta$, we have

$$f_\delta(\mathbf{x}) = \mathbb{E}_{\mathbf{u} \sim \mathbb{P}}[f(\mathbf{x} + \delta\mathbf{u})] = \frac{1}{\mathrm{Vol}(\mathbb{B}_1(\mathbf{0}))} \int_{\mathbf{u} \in \mathbb{B}_1(\mathbf{0})} f(\mathbf{x} + \delta\mathbf{u}) \, d\mathbf{u} = \frac{1}{\mathrm{Vol}(\mathbb{B}_\delta(\mathbf{0}))} \int_{\mathbf{v} \in \mathbb{B}_\delta(\mathbf{0})} f(\mathbf{x} + \mathbf{v}) \, d\mathbf{v}.$$

Since $f$ is $L$-Lipschitz, Bertsekas [9, Proposition 2.3] guarantees that $f_\delta$ is everywhere differentiable. Thus, we have $\nabla f_\delta(\mathbf{x})$ exists for any $\mathbf{x} \in \mathbb{R}^d$ and satisfies that

$$\lim_{\|\mathbf{h}\| \to 0} \frac{|f_\delta(\mathbf{x}+\mathbf{h}) - f_\delta(\mathbf{x}) - \langle \nabla f_\delta(\mathbf{x}), \mathbf{h} \rangle|}{\|\mathbf{h}\|} = 0. \tag{C.1}$$

Further, we have

$$\frac{f_\delta(\mathbf{x}+\mathbf{h}) - f_\delta(\mathbf{x})}{\|\mathbf{h}\|} = \frac{1}{\mathrm{Vol}(\mathbb{B}_\delta(\mathbf{0}))} \int_{\mathbf{v} \in \mathbb{B}_\delta(\mathbf{0})} \frac{f(\mathbf{x}+\mathbf{h}+\mathbf{v}) - f(\mathbf{x}+\mathbf{v})}{\|\mathbf{h}\|} \, d\mathbf{v}$$

Since $f$ is $L$-Lipschitz, we have $\frac{f(\mathbf{x}+\mathbf{h}+\mathbf{v}) - f(\mathbf{x}+\mathbf{v})}{\|\mathbf{h}\|} \leq L$. By the dominated convergence theorem, we have

$$\lim_{\|\mathbf{h}\| \to 0} \frac{f_\delta(\mathbf{x}+\mathbf{h}) - f_\delta(\mathbf{x})}{\|\mathbf{h}\|} = \frac{1}{\mathrm{Vol}(\mathbb{B}_\delta(\mathbf{0}))} \int_{\mathbf{v} \in \mathbb{B}_\delta(\mathbf{0})} \left( \lim_{\|\mathbf{h}\| \to 0} \frac{f(\mathbf{x}+\mathbf{h}+\mathbf{v}) - f(\mathbf{x}+\mathbf{v})}{\|\mathbf{h}\|} \right) d\mathbf{v}$$

Furthermore, Rademacher's theorem guarantees that $f$ is almost everywhere differentiable. Letting $U \subseteq \mathbb{B}_\delta(\mathbf{0})$ such that $\mathrm{Vol}(U) = \mathrm{Vol}(\mathbb{B}_\delta(\mathbf{0}))$ and $f$ is differentiable at $\mathbf{x} + \mathbf{v}$ for $\forall \mathbf{v} \in U$, we have

$$\lim_{\|\mathbf{h}\| \to 0} \frac{f_\delta(\mathbf{x}+\mathbf{h}) - f_\delta(\mathbf{x})}{\|\mathbf{h}\|} = \frac{1}{\mathrm{Vol}(U)} \int_{\mathbf{v} \in U} \left( \lim_{\|\mathbf{h}\| \to 0} \frac{f(\mathbf{x}+\mathbf{h}+\mathbf{v}) - f(\mathbf{x}+\mathbf{v})}{\|\mathbf{h}\|} \right) d\mathbf{v}, \tag{C.2}$$

and

$$\lim_{\|\mathbf{h}\| \to 0} \frac{|f(\mathbf{x}+\mathbf{h}+\mathbf{v}) - f(\mathbf{x}+\mathbf{v}) - \langle \nabla f(\mathbf{x}+\mathbf{v}), \mathbf{h} \rangle|}{\|\mathbf{h}\|} = 0. \tag{C.3}$$

Combining Eq. (C.1), Eq (C.2) and Eq. (C.3) together yields that

$$\lim_{\|\mathbf{h}\| \to 0} \frac{|\langle \nabla f_\delta(\mathbf{x}) - \mathbb{E}_{\mathbf{u} \sim \mathbb{P}}[\nabla f(\mathbf{x}+\delta \mathbf{u})], \mathbf{h} \rangle|}{\|\mathbf{h}\|} = 0.$$

Choosing $\mathbf{h} = t(\nabla f_\delta(\mathbf{x}) - \mathbb{E}_{\mathbf{u} \sim \mathbb{P}}[\nabla f(\mathbf{x} + \delta \mathbf{u})])$ with $t \to 0$, we have $\|\nabla f_\delta(\mathbf{x}) - \mathbb{E}_{\mathbf{u} \sim \mathbb{P}}[\nabla f(\mathbf{x} + \delta \mathbf{u})]\| = 0$.

It remains to show that $\nabla f_\delta(\mathbf{x}) \in \partial_\delta f(\mathbf{x})$ for any $\mathbf{x} \in \mathbb{R}^d$ using the proof argument by contradiction. In particular, we assume that there exists $\mathbf{x}_0 \in \mathbb{R}^d$ such that $\nabla f_\delta(\mathbf{x}_0) \notin \partial_\delta f(\mathbf{x}_0)$. Recall that

$$\partial_\delta f(\mathbf{x}_0) := \mathrm{conv}(\cup_{\mathbf{y} \in \mathbb{B}_\delta(\mathbf{x}_0)} \partial f(\mathbf{y})),$$

By the hyperplane separation theorem [74], there exists a unit vector $\mathbf{g} \in \mathbb{R}^d$ such that $\langle \mathbf{g}, \nabla f_\delta(\mathbf{x}_0) \rangle > 0$ and

$$\langle \mathbf{g}, \xi \rangle \leq 0, \quad \text{for any } \xi \in \cup_{\mathbf{y} \in \mathbb{B}_\delta(\mathbf{x}_0)} \partial f(\mathbf{y}). \tag{C.4}$$

However, we already obtain that $\nabla f_\delta(\mathbf{x}) = \mathbb{E}_{\mathbf{u} \sim \mathbb{P}}[\nabla f(\mathbf{x} + \delta \mathbf{u})]$ which implies that

$$\nabla f_\delta(\mathbf{x}_0) = \frac{1}{\mathrm{Vol}(\mathbb{B}_1(\mathbf{0}))} \int_{\mathbf{u} \in \mathbb{B}_1(\mathbf{0})} \nabla f(\mathbf{x}_0 + \delta \mathbf{u}) \, d\mathbf{u} = \frac{1}{\mathrm{Vol}(\mathbb{B}_\delta(\mathbf{0}))} \int_{\mathbf{y} \in \mathbb{B}_\delta(\mathbf{x}_0)} \nabla f(\mathbf{y}) \, d\mathbf{y}.$$

Thus, Eq. (C.4) implies that $\langle \mathbf{g}, \nabla f_\delta(\mathbf{x}_0) \rangle \leq 0$ which leads to a contradiction. Therefore, we conclude that $\nabla f_\delta(\mathbf{x}) \in \partial_\delta f(\mathbf{x})$ for any $\mathbf{x} \in \mathbb{R}^d$. This completes the proof.

# D  Missing Proofs for Gradient-Free Methods

In this section, we present some technical lemmas for analyzing the convergence property of gradient-free method and its two-phase version. We also give the proofs of Theorem 3.2 and 3.4.

## D.1  Technical lemmas

We provide two technical lemmas for analyzing Algorithm 1. The first lemma is a restatement of Shamir [75, Lemma 10] which gives an upper bound on the quantity $\mathbb{E}[\|\mathbf{g}^t\|^2 | \mathbf{x}^t]$ in terms of problem dimension $d \geq 1$ and the Lipschitz parameter $L > 0$. For the sake of completeness, we provide the proof details.

**Lemma D.1** *Suppose that $f$ is $L$-Lipschitz and let $\{\mathbf{g}^t\}_{t=0}^{T-1}$ and $\{\mathbf{x}^t\}_{t=0}^{T-1}$ be generated by Algorithm 1. Then, we have $\mathbb{E}[\mathbf{g}^t | \mathbf{x}^t] = \nabla f_\delta(\mathbf{x}^t)$ and $\mathbb{E}[\|\mathbf{g}^t\|^2 | \mathbf{x}^t] \leq 16\sqrt{2\pi} d L^2$.*

*Proof.* By the definition of $\mathbf{g}^t$ and the symmetry of the distribution of $\mathbf{w}^t$, we have

$$\begin{aligned}
\mathbb{E}[\mathbf{g}^t \mid \mathbf{x}^t] &= \mathbb{E}\left[\tfrac{d}{2\delta}(f(\mathbf{x}^t + \delta\mathbf{w}^t) - f(\mathbf{x}^t - \delta\mathbf{w}^t))\mathbf{w}^t \mid \mathbf{x}^t\right] \\
&= \tfrac{1}{2}\left(\mathbb{E}\left[\tfrac{d}{\delta}f(\mathbf{x}^t + \delta\mathbf{w}^t)\mathbf{w}^t \mid \mathbf{x}^t\right] + \mathbb{E}\left[\tfrac{d}{\delta}f(\mathbf{x}^t + \delta(-\mathbf{w}^t))(-\mathbf{w}^t) \mid \mathbf{x}^t\right]\right) \\
&= \tfrac{1}{2}\left(\nabla f_\delta(\mathbf{x}^t) + \nabla f_\delta(\mathbf{x}^t)\right) = \nabla f_\delta(\mathbf{x}^t).
\end{aligned}$$

It remains to show that $\mathbb{E}[\|\mathbf{g}^t\|^2 \mid \mathbf{x}^t] \leq 16\sqrt{2\pi}dL^2$. Indeed, since $\|\mathbf{w}^t\| = 1$, we have

$$\mathbb{E}[\|\mathbf{g}^t\|^2 \mid \mathbf{x}^t] = \mathbb{E}\left[\tfrac{d^2}{4\delta^2}(f(\mathbf{x}^t + \delta\mathbf{w}^t) - f(\mathbf{x}^t - \delta\mathbf{w}^t))^2\|\mathbf{w}^t\|^2 \mid \mathbf{x}^t\right] \leq \mathbb{E}\left[\tfrac{d^2}{4\delta^2}(f(\mathbf{x}^t + \delta\mathbf{w}^t) - f(\mathbf{x}^t - \delta\mathbf{w}^t))^2 \mid \mathbf{x}^t\right].$$

Using the elementary inequality $(a-b)^2 \leq 2a^2 + 2b^2$, we have

$$\begin{aligned}
&\mathbb{E}[(f(\mathbf{x}^t + \delta\mathbf{w}^t) - f(\mathbf{x}^t - \delta\mathbf{w}^t))^2 \mid \mathbf{x}^t] \\
&= \mathbb{E}[(f(\mathbf{x}^t + \delta\mathbf{w}^t) - \mathbb{E}[f(\mathbf{x}^t + \delta\mathbf{w}^t) \mid \mathbf{x}^t] - (f(\mathbf{x}^t - \delta\mathbf{w}^t) - \mathbb{E}[f(\mathbf{x}^t + \delta\mathbf{w}^t) \mid \mathbf{x}^t]))^2 \mid \mathbf{x}^t] \\
&\leq 2\mathbb{E}[(f(\mathbf{x}^t + \delta\mathbf{w}^t) - \mathbb{E}[f(\mathbf{x}^t + \delta\mathbf{w}^t) \mid \mathbf{x}^t])^2 \mid \mathbf{x}^t] + 2\mathbb{E}[(f(\mathbf{x}^t - \delta\mathbf{w}^t) - \mathbb{E}[f(\mathbf{x}^t + \delta\mathbf{w}^t) \mid \mathbf{x}^t])^2 \mid \mathbf{x}^t].
\end{aligned}$$

Since $\mathbf{w}^t$ has a symmetric distribution around the origin, we have

$$\mathbb{E}[(f(\mathbf{x}^t + \delta\mathbf{w}^t) - \mathbb{E}[f(\mathbf{x}^t + \delta\mathbf{w}^t) \mid \mathbf{x}^t])^2 \mid \mathbf{x}^t] = \mathbb{E}[(f(\mathbf{x}^t - \delta\mathbf{w}^t) - \mathbb{E}[f(\mathbf{x}^t + \delta\mathbf{w}^t) \mid \mathbf{x}^t])^2 \mid \mathbf{x}^t].$$

Putting these pieces together yields that

$$\mathbb{E}[\|\mathbf{g}^t\|^2 \mid \mathbf{x}^t] \leq \tfrac{d^2}{\delta^2}\mathbb{E}[(f(\mathbf{x}^t + \delta\mathbf{w}^t) - \mathbb{E}[f(\mathbf{x}^t + \delta\mathbf{w}^t) \mid \mathbf{x}^t])^2 \mid \mathbf{x}^t]. \tag{D.1}$$

For simplicity, we let $h(\mathbf{w}) = f(\mathbf{x}^t + \delta\mathbf{w})$. Since $f$ is $L$-Lipschitz, this function $h$ is $\delta L$-Lipschitz given a fixed $\mathbf{x}^t$. In addition, $\mathbf{w}^t \in \mathbb{R}^d$ is sampled uniformly from a unit sphere. Then, by Wainwright [81, Proposition 3.11 and Example 3.12], we have

$$\mathbb{P}(|h(\mathbf{w}^t) - \mathbb{E}[h(\mathbf{w}^t)]| \geq \alpha) \leq 2\sqrt{2\pi}e^{-\frac{\alpha^2 d}{8\delta^2 L^2}}.$$

Then, we have

$$\begin{aligned}
\mathbb{E}[(h(\mathbf{w}^t) - \mathbb{E}[h(\mathbf{w}^t)])^2] &= \int_0^{+\infty} \mathbb{P}((h(\mathbf{w}^t) - \mathbb{E}[h(\mathbf{w}^t)])^2 \geq \alpha) \, d\alpha \\
&= \int_0^{+\infty} \mathbb{P}(|h(\mathbf{w}^t) - \mathbb{E}[h(\mathbf{w}^t)]| \geq \sqrt{\alpha}) \, d\alpha \leq 2\sqrt{2\pi} \int_0^{+\infty} e^{-\frac{\alpha d}{8\delta^2 L^2}} \, d\alpha \\
&= 2\sqrt{2\pi} \cdot \tfrac{8\delta^2 L^2}{d} = \tfrac{16\sqrt{2\pi}\delta^2 L^2}{d}.
\end{aligned}$$

By the definition of $h$, we have

$$\mathbb{E}[(f(\mathbf{x}^t + \delta\mathbf{w}^t) - \mathbb{E}[f(\mathbf{x}^t + \delta\mathbf{w}^t) \mid \mathbf{x}^t])^2 \mid \mathbf{x}^t] \leq \tfrac{16\sqrt{2\pi}\delta^2 L^2}{d}. \tag{D.2}$$

Combining Eq. (D.1) and Eq. (D.2) yields the desired inequality. $\qquad\square$

The second lemma gives a key descent inequality for analyzing Algorithm 1.

**Lemma D.2** *Suppose that $f$ is $L$-Lipschitz and let $\{\mathbf{x}^t\}_{t=0}^{T-1}$ be generated by Algorithm 1. Then, we have*

$$\mathbb{E}[\|\nabla f_\delta(\mathbf{x}^t)\|^2] \leq \tfrac{\mathbb{E}[f_\delta(\mathbf{x}^t)] - \mathbb{E}[f_\delta(\mathbf{x}^{t+1})]}{\eta} + \eta \cdot \tfrac{(8\sqrt{2\pi})cd^{3/2}L^3}{\delta}, \quad \text{for all } 0 \leq t \leq T-1.$$

*where $c > 0$ is a constant appearing in the smoothing parameter of $f_\delta$ (cf. Proposition 2.3).*

*Proof.* By Proposition 2.3, we have $f_\delta$ is differentiable and $L$-Lipschitz with the $\frac{cL\sqrt{d}}{\delta}$-Lipschitz gradient where $c > 0$ is a constant. This implies that

$$f_\delta(\mathbf{x}^{t+1}) \leq f_\delta(\mathbf{x}^t) - \eta\langle\nabla f_\delta(\mathbf{x}^t), \mathbf{g}^t\rangle + \tfrac{c\eta^2 L\sqrt{d}}{2\delta}\|\mathbf{g}^t\|^2.$$

Taking the expectation of both sides conditioned on $\mathbf{x}^t$ and using Lemma D.1, we have

$$\begin{aligned}
\mathbb{E}[f_\delta(\mathbf{x}^{t+1}) \mid \mathbf{x}^t] &\leq f_\delta(\mathbf{x}^t) - \eta\langle\nabla f_\delta(\mathbf{x}^t), \mathbb{E}[\mathbf{g}^t \mid \mathbf{x}^t]\rangle + \tfrac{c\eta^2 L\sqrt{d}}{2\delta}\mathbb{E}[\|\mathbf{g}^t\|^2 \mid \mathbf{x}^t] \\
&\leq f_\delta(\mathbf{x}^t) - \eta\|\nabla f_\delta(\mathbf{x}^t)\|^2 + \eta^2 \cdot \tfrac{cL\sqrt{d}}{2\delta} \cdot 16\sqrt{2\pi}dL^2 \\
&= f_\delta(\mathbf{x}^t) - \eta\|\nabla f_\delta(\mathbf{x}^t)\|^2 + \eta^2 \cdot (8\sqrt{2\pi})cd^{3/2}L^3\delta^{-1}.
\end{aligned}$$

Taking the expectation of both sides and rearranging yields that

$$\mathbb{E}[\|\nabla f_\delta(\mathbf{x}^t)\|^2] \leq \frac{\mathbb{E}[f_\delta(\mathbf{x}^t)] - \mathbb{E}[f_\delta(\mathbf{x}^{t+1})]}{\eta} + \eta \cdot \frac{(8\sqrt{2\pi})cd^{3/2}L^3}{\delta}.$$

This completes the proof. $\qquad\square$

We also present a proposition which is crucial to deriving the large deviation property of Algorithm 2.

**Proposition D.3** *Suppose that $\Omega$ is a polish space with a Borel probability measure $\mathbb{P}$ and let $\{\emptyset, \Omega\} = \mathcal{F}_0 \subseteq \mathcal{F}_1 \subseteq \mathcal{F}_2 \subseteq \ldots$ be a sequence of filtration. For an integer $N \geq 1$, we define a martingale-difference sequence of Borel functions $\{\zeta_k\}_{k=1}^N \subseteq \mathbb{R}^n$ such that $\zeta_k$ is $\mathcal{F}_k$-measurable and $\mathbb{E}[\zeta_k \mid \mathcal{F}_{k-1}] = 0$. Then, if $\mathbb{E}[\|\zeta_k\|^2] \leq \sigma_k^2$ for all $k \geq 1$, we have $\mathbb{E}[\|\sum_{k=1}^N \zeta_k\|^2] \leq \sum_{k=1}^N \sigma_k^2$ and the following statement holds true,*

$$\text{Prob}\left(\left\|\sum_{k=1}^N \zeta_k\right\|^2 \geq \lambda \sum_{k=1}^N \sigma_k^2\right) \leq \tfrac{1}{\lambda}, \quad \text{for all } \lambda \geq 0.$$

This is a general result concerning about the large deviations of vector martingales; see, e.g., Juditsky and Nemirovski [54, Theorem 2.1] or Ghadimi and Lan [44, Lemma 2.3].

## D.2 Proof of Theorem 3.2

Summing up the inequality in Lemma D.2 over $t = 0, 1, 2, \ldots, T - 1$ yields that

$$\sum_{t=0}^{T-1} \mathbb{E}[\|\nabla f_\delta(\mathbf{x}^t)\|^2] \leq \frac{f_\delta(\mathbf{x}^0) - \mathbb{E}[f_\delta(\mathbf{x}^T)]}{\eta} + \eta \cdot \frac{(8\sqrt{2\pi})cd^{3/2}L^3 T}{\delta}.$$

By Proposition 2.3, we have $f(\mathbf{x}_0) \leq f_\delta(\mathbf{x}_0) \leq f(\mathbf{x}_0) + \delta L$. In addition, we see from the definition of $f_\delta$ that $f_\delta(\mathbf{x}) \geq \inf_{\mathbf{x} \in \mathbb{R}^d} f(\mathbf{x})$ for any $\mathbf{x} \in \mathbb{R}^d$ and thus $\mathbb{E}[f_\delta(\mathbf{x}^T)] \geq \inf_{\mathbf{x} \in \mathbb{R}^d} f(\mathbf{x})$. Putting these pieces together with $f \in \mathcal{F}_d(\Delta, L)$ yields that

$$f_\delta(\mathbf{x}^0) - \mathbb{E}[f_\delta(\mathbf{x}^T)] \leq f(\mathbf{x}_0) - \inf_{\mathbf{x} \in \mathbb{R}^d} f(\mathbf{x}) + \delta L \leq \Delta + \delta L.$$

Therefore, we conclude that

$$\frac{1}{T}\left(\sum_{t=0}^{T-1} \mathbb{E}[\|\nabla f_\delta(\mathbf{x}^t)\|^2]\right) \leq \frac{\Delta + \delta L}{\eta T} + \eta \cdot \frac{(8\sqrt{2\pi})cd^{3/2}L^3}{\delta} \leq \frac{\Delta + \delta L}{\eta T} + \eta \cdot \frac{100cd^{3/2}L^3}{\delta}.$$

Recalling that $\eta = \frac{1}{10}\sqrt{\frac{\delta(\Delta + \delta L)}{cd^{3/2}L^3 T}}$, we have

$$\frac{1}{T}\left(\sum_{t=0}^{T-1} \mathbb{E}[\|\nabla f_\delta(\mathbf{x}^t)\|^2]\right) \leq 20\sqrt{\frac{cd^{3/2}L^3}{T}(L + \tfrac{\Delta}{\delta})}.$$

Since the random count $R \in \{0, 1, 2, \ldots, T - 1\}$ is uniformly sampled, we have

$$\mathbb{E}[\|\nabla f_\delta(\mathbf{x}^R)\|^2] = \frac{1}{T}\left(\sum_{t=0}^{T-1} \mathbb{E}[\|\nabla f_\delta(\mathbf{x}^t)\|^2]\right) \leq 20\sqrt{\frac{cd^{3/2}L^3}{T}(L + \tfrac{\Delta}{\delta})}. \tag{D.3}$$

By Theorem 3.1, we have $\nabla f_\delta(\mathbf{x}^R) \in \partial_\delta f(\mathbf{x}^R)$. This together with the above inequality implies that

$$\mathbb{E}[\min\{\|\mathbf{g}\| : \mathbf{g} \in \partial_\delta f(\mathbf{x}^R)\}] \leq \mathbb{E}[\|\nabla f_\delta(\mathbf{x}^R)\|] \leq 5\left(\frac{cd^{3/2}L^3}{T}(L + \tfrac{\Delta}{\delta})\right)^{\frac{1}{4}}.$$

Therefore, we conclude that there exists some $T > 0$ such that the output of Algorithm 1 satisfies that $\mathbb{E}[\min\{\|\mathbf{g}\| : \mathbf{g} \in \partial_\delta f(\mathbf{x}^R)\}] \leq \epsilon$ and the total number of calling the function value oracles is bounded by

$$O\left(d^{\frac{3}{2}}\left(\frac{L^4}{\epsilon^4} + \frac{\Delta L^3}{\delta\epsilon^4}\right)\right).$$

This completes the proof.

### D.3 Proof of Theorem 3.4

By the definition of $s^\star$ and using the Cauchy -Schwarz inequality, we have

$$\|\mathbf{g}_{s^\star}\|^2 = \min_{s=0,1,2,\dots,S-1}\|\mathbf{g}_s\|^2 \leq \min_{s=0,1,2,\dots,S-1}\left\{2\|\nabla f_\delta(\bar{\mathbf{x}}_s)\|^2 + 2\|\mathbf{g}_s - \nabla f_\delta(\bar{\mathbf{x}}_s)\|^2\right\} \quad \text{(D.4)}$$

$$\leq 2\left(\min_{s=0,1,2,\dots,S-1}\|\nabla f_\delta(\bar{\mathbf{x}}_s)\|^2 + \max_{s=0,1,2,\dots,S-1}\|\mathbf{g}_s - \nabla f_\delta(\bar{\mathbf{x}}_s)\|^2\right).$$

This implies that

$$\|\nabla f_\delta(\bar{\mathbf{x}}_{s^\star})\|^2 \leq 2\|\mathbf{g}_{s^\star}\|^2 + 2\|\mathbf{g}_{s^\star} - \nabla f_\delta(\bar{\mathbf{x}}_{s^\star})\|^2 \quad \text{(D.5)}$$

$$\overset{\text{Eq. (D.4)}}{\leq} 4\left(\min_{s=0,1,2,\dots,S-1}\|\nabla f_\delta(\bar{\mathbf{x}}_s)\|^2\right) + 4\left(\max_{s=0,1,2,\dots,S-1}\|\mathbf{g}_s - \nabla f_\delta(\bar{\mathbf{x}}_s)\|^2\right) + 2\|\mathbf{g}_{s^\star} - \nabla f_\delta(\bar{\mathbf{x}}_{s^\star})\|^2.$$

The next step is to provide the probabilistic bounds on all the terms in the right-hand side of Eq. (D.5). In particular, for each $s = 0, 1, 2, \dots, S-1$, we have $\bar{\mathbf{x}}_s$ is an output obtained by calling Algorithm 1 with $\mathbf{x}^0$, $d$, $\delta$, $T$ and $\eta = \frac{1}{10}\sqrt{\frac{\delta(\Delta+\delta L)}{cd^{3/2}L^3 T}}$. Then, Eq. (D.3) in the proof of Theorem 3.2 implies that

$$\mathbb{E}[\|\nabla f_\delta(\bar{\mathbf{x}}_s)\|^2] \leq 20\sqrt{\frac{cd^{3/2}L^3}{T}\left(L + \frac{\Delta}{\delta}\right)}.$$

Using the Markov's inequality, we have

$$\mathrm{Prob}\left(\|\nabla f_\delta(\bar{\mathbf{x}}_s)\|^2 \geq 40\sqrt{\frac{cd^{3/2}L^3}{T}\left(L + \frac{\Delta}{\delta}\right)}\right) \leq \tfrac{1}{2}.$$

Thus, we have

$$\mathrm{Prob}\left(\min_{s=0,1,2,\dots,S-1}\|\nabla f_\delta(\bar{\mathbf{x}}_s)\|^2 \geq 40\sqrt{\frac{cd^{3/2}L^3}{T}\left(L + \frac{\Delta}{\delta}\right)}\right) \leq 2^{-S}. \quad \text{(D.6)}$$

Further, for each $s = 0, 1, 2, \dots, S-1$, we have

$$\mathbf{g}_s - \nabla f_\delta(\bar{\mathbf{x}}_s) = \tfrac{1}{B}\sum_{k=0}^{B-1}(\mathbf{g}_s^k - \nabla f_\delta(\bar{\mathbf{x}}_s)).$$

By Lemma D.1, we have $\mathbb{E}[\mathbf{g}_s^t|\bar{\mathbf{x}}_s] = \nabla f_\delta(\bar{\mathbf{x}}_s)$ and $\mathbb{E}[\|\mathbf{g}_s^t\|^2|\bar{\mathbf{x}}_s] \leq 16\sqrt{2\pi}dL^2$. This implies that

$$\mathbb{E}[\mathbf{g}_s^t - \nabla f_\delta(\bar{\mathbf{x}}_s)|\bar{\mathbf{x}}_s] = 0, \quad \mathbb{E}[\|\mathbf{g}_s^t - \nabla f_\delta(\bar{\mathbf{x}}_s)\|^2] \leq 16\sqrt{2\pi}dL^2.$$

This together with Proposition D.3 yields that

$$\mathrm{Prob}\left(\|\mathbf{g}_s - \nabla f_\delta(\bar{\mathbf{x}}_s)\|^2 \geq \tfrac{\lambda(16\sqrt{2\pi}dL^2)}{B}\right) = \mathrm{Prob}\left(\left\|\sum_{k=0}^{B-1}(\mathbf{g}_s^k - \nabla f_\delta(\bar{\mathbf{x}}_s))\right\|^2 \geq \lambda B(16\sqrt{2\pi}dL^2)\right) \leq \tfrac{1}{\lambda}.$$

Therefore, we conclude that

$$\mathrm{Prob}\left(\max_{s=0,1,2,\dots,S-1}\|\mathbf{g}_s - \nabla f_\delta(\bar{\mathbf{x}}_s)\|^2 \geq \tfrac{\lambda(16\sqrt{2\pi}dL^2)}{B}\right) \leq \tfrac{S}{\lambda}. \quad \text{(D.7)}$$

By the similar argument, we have

$$\mathrm{Prob}(\|\mathbf{g}_{s^\star} - \nabla f_\delta(\bar{\mathbf{x}}_{s^\star})\|^2 \geq \tfrac{\lambda(16\sqrt{2\pi}dL^2)}{B}) \leq \tfrac{1}{\lambda}. \quad \text{(D.8)}$$

Combining Eq. (D.5), Eq. (D.6), Eq. (D.7) and Eq. (D.8) yields that

$$\mathrm{Prob}\left(\|\nabla f_\delta(\bar{\mathbf{x}}_{s^\star})\|^2 \geq 160\sqrt{\frac{cd^{3/2}L^3}{T}\left(L + \frac{\Delta}{\delta}\right)} + \tfrac{\lambda(96\sqrt{2\pi}dL^2)}{B}\right) \leq \tfrac{S+1}{\lambda} + 2^{-S}, \quad \text{for all } \lambda > 0. \quad \text{(D.9)}$$

We set $\lambda = \frac{2(S+1)}{\Lambda}$ and the parameters $(T, S, B)$ as follows,

$$T = cd^{3/2}L^3\left(L + \tfrac{\Delta}{\delta}\right)\left(\tfrac{160}{\epsilon^2}\right)^2, \quad S = \lceil\log_2(\tfrac{2}{\Lambda})\rceil, \quad B = \tfrac{(384\sqrt{2\pi}dL^2)(S+1)}{\Lambda\epsilon^2}.$$

Then, we have

$$\text{Prob}\left(\|\nabla f_\delta(\bar{\mathbf{x}}_{s^\star})\|^2 \geq \epsilon^2\right) \leq \text{Prob}\left(\|\nabla f_\delta(\bar{\mathbf{x}}_{s^\star})\|^2 \geq 160\sqrt{\frac{cd^{3/2}L^3}{T}(L+\frac{\Delta}{\delta})} + \frac{\lambda(96\sqrt{2\pi}dL^2)}{B}\right) \leq \Lambda.$$

By Theorem 3.1, we have $\nabla f_\delta(\bar{\mathbf{x}}_{s^\star}) \in \partial_\delta f(\bar{\mathbf{x}}_{s^\star})$. This together with the above inequality implies that there exists some $T, S, B > 0$ such that the output of Algorithm 2 satisfies that $\mathbb{E}[\min\{\|\mathbf{g}\| : \mathbf{g} \in \partial_\delta f(\bar{\mathbf{x}}_{s^\star})\}] \leq \epsilon$ and the total number of calling the function value oracles is bounded by

$$O\left(d^{\frac{3}{2}}\left(\frac{L^4}{\epsilon^4} + \frac{\Delta L^3}{\delta\epsilon^4}\right)\log_2\left(\frac{1}{\Lambda}\right) + \frac{dL^2}{\Lambda\epsilon^2}\log_2\left(\frac{1}{\Lambda}\right)\right).$$

This completes the proof.

# E    Missing Proofs for Stochastic Gradient-Free Methods

In this section, we present some technical lemmas for analyzing the convergence property of stochastic gradient-free method and its two-phase version. We also give the proofs of Theorem 3.5 and 3.6.

## E.1    Technical lemmas

We provide two technical lemmas for analyzing Algorithm 3. The first lemma gives an upper bound on the quantity $\mathbb{E}[\|\hat{\mathbf{g}}^t\|^2|\mathbf{x}^t]$ in terms of problem dimension $d \geq 1$ and the constant $G > 0$. The proof is based on a modification of the proof of Lemma D.1.

**Lemma E.1** *Suppose that $\{\hat{\mathbf{g}}^t\}_{t=0}^{T-1}$ and $\{\mathbf{x}^t\}_{t=0}^{T-1}$ are generated by Algorithm 3. Then, we have $\mathbb{E}[\hat{\mathbf{g}}^t|\mathbf{x}^t] = \nabla f_\delta(\mathbf{x}^t)$ and $\mathbb{E}[\|\hat{\mathbf{g}}^t\|^2|\mathbf{x}^t] \leq 16\sqrt{2\pi}dG^2$.*

*Proof.* By the definition of $\hat{\mathbf{g}}^t$ and the symmetry of the distribution of $\mathbf{w}^t$, we have

$$\begin{aligned}
\mathbb{E}[\hat{\mathbf{g}}^t \mid \mathbf{x}^t] &= \mathbb{E}\left[\tfrac{d}{2\delta}(F(\mathbf{x}^t + \delta\mathbf{w}^t, \xi^t) - F(\mathbf{x}^t - \delta\mathbf{w}^t, \xi^t))\mathbf{w}^t \mid \mathbf{x}^t\right] \\
&= \tfrac{1}{2}\left(\mathbb{E}\left[\tfrac{d}{\delta}F(\mathbf{x}^t + \delta\mathbf{w}^t, \xi^t)\mathbf{w}^t \mid \mathbf{x}^t\right] + \mathbb{E}\left[\tfrac{d}{\delta}F(\mathbf{x}^t + \delta(-\mathbf{w}^t), \xi^t)(-\mathbf{w}^t) \mid \mathbf{x}^t\right]\right) \\
&= \mathbb{E}\left[\tfrac{d}{\delta}F(\mathbf{x}^t + \delta\mathbf{w}^t, \xi^t)\mathbf{w}^t \mid \mathbf{x}^t\right].
\end{aligned}$$

By the tower property, we have

$$\mathbb{E}[\hat{\mathbf{g}}^t \mid \mathbf{x}^t] = \mathbb{E}\left[\tfrac{d}{\delta}\mathbb{E}[F(\mathbf{x}^t + \delta\mathbf{w}^t, \xi^t)\mathbf{w}^t \mid \mathbf{x}^t, \mathbf{w}^t] \mid \mathbf{x}^t\right] = \mathbb{E}\left[\tfrac{d}{\delta}f(\mathbf{x}^t + \delta\mathbf{w}^t)\mathbf{w}^t \mid \mathbf{x}^t\right] = \nabla f_\delta(\mathbf{x}^t).$$

It remains to show that $\mathbb{E}[\|\hat{\mathbf{g}}^t\|^2 \mid \mathbf{x}^t] \leq 16\sqrt{2\pi}dG^2$. Indeed, by using the same argument as used in the proof of Lemma D.1, we have

$$\mathbb{E}[\|\hat{\mathbf{g}}^t\|^2|\mathbf{x}^t] \leq \tfrac{d^2}{\delta^2}\mathbb{E}[(F(\mathbf{x}^t + \delta\mathbf{w}^t, \xi^t) - \mathbb{E}[F(\mathbf{x}^t + \delta\mathbf{w}^t, \xi^t) \mid \mathbf{x}^t, \xi^t])^2 \mid \mathbf{x}^t]. \tag{E.1}$$

For simplicity, we let $h(\mathbf{w}) = F(\mathbf{x}^t + \delta\mathbf{w}, \xi^t)$. Since $F(\cdot, \xi)$ is $L(\xi)$-Lipschitz, this function $h$ is $\delta L(\xi^t)$-Lipschitz given a fixed $\mathbf{x}^t$ and $\xi^t$. In addition, $\mathbf{w}^t \in \mathbb{R}^d$ is sampled uniformly from a unit sphere. Then, by Wainwright [81, Proposition 3.11 and Example 3.12], we have

$$\mathbb{P}(|h(\mathbf{w}^t) - \mathbb{E}[h(\mathbf{w}^t)]| \geq \alpha) \leq 2\sqrt{2\pi}e^{-\frac{\alpha^2 d}{8\delta^2 L(\xi^t)^2}}.$$

Then, we have

$$\begin{aligned}
\mathbb{E}[(h(\mathbf{w}^t) - \mathbb{E}[h(\mathbf{w}^t)])^2] &= \int_0^{+\infty} \mathbb{P}((h(\mathbf{w}^t) - \mathbb{E}[h(\mathbf{w}^t)])^2 \geq \alpha)\, d\alpha \\
&= \int_0^{+\infty} \mathbb{P}(|h(\mathbf{w}^t) - \mathbb{E}[h(\mathbf{w}^t)]| \geq \sqrt{\alpha})\, d\alpha \leq 2\sqrt{2\pi}\int_0^{+\infty} e^{-\frac{\alpha d}{8\delta^2 L(\xi^t)^2}}\, d\alpha \\
&= 2\sqrt{2\pi} \cdot \tfrac{8\delta^2 L(\xi^t)^2}{d} = \tfrac{16\sqrt{2\pi}\delta^2 L(\xi^t)^2}{d}.
\end{aligned}$$

By the definition of $h$, we have

$$\mathbb{E}[(F(\mathbf{x}^t + \delta\mathbf{w}^t, \xi^t) - \mathbb{E}[F(\mathbf{x}^t + \delta\mathbf{w}^t, \xi^t) \mid \mathbf{x}^t, \xi^t])^2 \mid \mathbf{x}^t] \leq \tfrac{16\sqrt{2\pi}\delta^2}{d}\mathbb{E}[L(\xi^t)^2].$$

Since $\xi^t$ is simulated from the distribution $\mathbb{P}_\mu$, we have $\mathbb{E}[L(\xi^t)^2] \leq G^2$. Plugging this into the above inequality, we have

$$\mathbb{E}[(F(\mathbf{x}^t + \delta \mathbf{w}^t, \xi^t) - \mathbb{E}[F(\mathbf{x}^t + \delta \mathbf{w}^t, \xi^t) \mid \mathbf{x}^t, \xi^t])^2 \mid \mathbf{x}^t] \leq \tfrac{16\sqrt{2\pi}\delta^2 G^2}{d} \qquad \text{(E.2)}$$

Combining Eq. (E.1) and Eq. (E.2) yields the desired inequality. $\qquad\square$

The second lemma gives a key descent inequality for analyzing Algorithm 3.

**Lemma E.2** *Suppose that $\{\mathbf{x}^t\}_{t=0}^{T-1}$ are generated by Algorithm 3. Then, we have*

$$\mathbb{E}[\|\nabla f_\delta(\mathbf{x}^t)\|^2] \leq \frac{\mathbb{E}[f_\delta(\mathbf{x}^t)] - \mathbb{E}[f_\delta(\mathbf{x}^{t+1})]}{\eta} + \eta \cdot \frac{(8\sqrt{2\pi})cd^{3/2}G^3}{\delta}, \quad \text{for all } 0 \leq t \leq T-1.$$

*Proof.* Since $f(\cdot) = \mathbb{E}_{\xi \in \mathbb{P}_\mu}[F(\cdot, \xi)]$ and $F(\cdot, \xi)$ is $L(\xi)$-Lipschitz with $\mathbb{E}_{\xi \in \mathbb{P}_\mu}[L^2(\xi)] \leq G^2$ for some $G > 0$, we have $f$ is $G$-Lipschitz. Then, by Proposition 2.3, we have $f_\delta$ is differentiable with the $\frac{cG\sqrt{d}}{\delta}$-Lipschitz gradient where $c > 0$ is a constant. This implies that

$$f_\delta(\mathbf{x}^{t+1}) \leq f_\delta(\mathbf{x}^t) - \eta \langle \nabla f_\delta(\mathbf{x}^t), \hat{\mathbf{g}}^t \rangle + \frac{c\eta^2 G\sqrt{d}}{2\delta}\|\hat{\mathbf{g}}^t\|^2.$$

Taking the expectation of both sides conditioned on $\mathbf{x}^t$ and using Lemma E.1, we have

$$
\begin{aligned}
\mathbb{E}[f_\delta(\mathbf{x}^{t+1}) \mid \mathbf{x}^t] &\leq f_\delta(\mathbf{x}^t) - \eta \langle \nabla f_\delta(\mathbf{x}^t), \mathbb{E}[\hat{\mathbf{g}}^t \mid \mathbf{x}^t] \rangle + \tfrac{c\eta^2 G\sqrt{d}}{2\delta}\mathbb{E}[\|\hat{\mathbf{g}}^t\|^2 \mid \mathbf{x}^t] \\
&\leq f_\delta(\mathbf{x}^t) - \eta \|\nabla f_\delta(\mathbf{x}^t)\|^2 + \eta^2 \cdot \tfrac{cG\sqrt{d}}{2\delta} \cdot 16\sqrt{2\pi}dG^2 \\
&= f_\delta(\mathbf{x}^t) - \eta \|\nabla f_\delta(\mathbf{x}^t)\|^2 + \eta^2 \cdot \tfrac{(8\sqrt{2\pi})cd^{3/2}G^3}{\delta}.
\end{aligned}
$$

Taking the expectation of both sides and rearranging yields that

$$\mathbb{E}[\|\nabla f_\delta(\mathbf{x}^t)\|^2] \leq \frac{\mathbb{E}[f_\delta(\mathbf{x}^t)] - \mathbb{E}[f_\delta(\mathbf{x}^{t+1})]}{\eta} + \eta \cdot \frac{(8\sqrt{2\pi})cd^{3/2}G^3}{\delta}.$$

This completes the proof. $\qquad\square$

### E.2  Proof of Theorem 3.5

Summing up the inequality in Lemma E.2 over $t = 0, 1, 2, \ldots, T-1$ yields that

$$\sum_{t=0}^{T-1} \mathbb{E}[\|\nabla f_\delta(\mathbf{x}^t)\|^2] \leq \frac{f_\delta(\mathbf{x}^0) - \mathbb{E}[f_\delta(\mathbf{x}^T)]}{\eta} + \eta \cdot \frac{(8\sqrt{2\pi})cd^{3/2}G^3 T}{\delta}.$$

Since $f(\cdot) = \mathbb{E}_{\xi \in \mathbb{P}_\mu}[F(\cdot, \xi)]$ and $F(\cdot, \xi)$ is $L(\xi)$-Lipschitz with $\mathbb{E}_{\xi \in \mathbb{P}_\mu}[L^2(\xi)] \leq G^2$ for some $G > 0$, we have $f$ is $G$-Lipschitz. Thus, we have $f \in \mathcal{F}_d(\Delta, L)$. By using the same argument as used in the proof of Theorem 3.2, we have

$$\frac{1}{T}\left(\sum_{t=0}^{T-1} \mathbb{E}[\|\nabla f_\delta(\mathbf{x}^t)\|^2]\right) \leq \frac{\Delta + \delta G}{\eta T} + \eta \cdot \frac{(8\sqrt{2\pi})cd^{3/2}G^3}{\delta} \leq \frac{\Delta + \delta G}{\eta T} + \eta \cdot \frac{100cd^{3/2}G^3}{\delta}.$$

Recalling that $\eta = \frac{1}{10}\sqrt{\frac{\delta(\Delta + \delta G)}{cd^{3/2}G^3 T}}$, we have

$$\frac{1}{T}\left(\sum_{t=0}^{T-1} \mathbb{E}[\|\nabla f_\delta(\mathbf{x}^t)\|^2]\right) \leq 20\sqrt{\frac{cd^{3/2}G^3}{T}(G + \tfrac{\Delta}{\delta})}.$$

Since the random count $R \in \{0, 1, 2, \ldots, T-1\}$ is uniformly sampled, we have

$$\mathbb{E}[\|\nabla f_\delta(\mathbf{x}^R)\|^2] = \frac{1}{T}\left(\sum_{t=0}^{T-1} \mathbb{E}[\|\nabla f_\delta(\mathbf{x}^t)\|^2]\right) \leq 20\sqrt{\frac{cd^{3/2}G^3}{T}(G + \tfrac{\Delta}{\delta})}. \qquad \text{(E.3)}$$

By Theorem 3.1, we have $\nabla f_\delta(\mathbf{x}^R) \in \partial_\delta f(\mathbf{x}^R)$. This together with the above inequality implies that

$$\mathbb{E}[\min\{\|\mathbf{g}\| : \mathbf{g} \in \partial_\delta f(\mathbf{x}^R)\}] \leq \mathbb{E}[\|\nabla f_\delta(\mathbf{x}^R)\|] \leq 5\left(\frac{cd^{3/2}G^3}{T}(G + \tfrac{\Delta}{\delta})\right)^{\frac{1}{4}}.$$

Therefore, we conclude that there exists some $T > 0$ such that the output of Algorithm 3 satisfies that $\mathbb{E}[\min\{\|\mathbf{g}\| : \mathbf{g} \in \partial_\delta f(\mathbf{x}^R)\}] \leq \epsilon$ and the total number of calling the function value oracles is bounded by

$$O\left(d^{\frac{3}{2}}\left(\frac{G^4}{\epsilon^4} + \frac{\Delta G^3}{\delta\epsilon^4}\right)\right).$$

This completes the proof.

### E.3    Proof of Theorem 3.6

By the definition of $s^\star$ and using the Cauchy -Schwarz inequality, we have

$$\|\hat{\mathbf{g}}_{s^\star}\|^2 = \min_{s=0,1,2,\ldots,S-1}\|\hat{\mathbf{g}}_s\|^2 \leq \min_{s=0,1,2,\ldots,S-1}\left\{2\|\nabla f_\delta(\bar{\mathbf{x}}_s)\|^2 + 2\|\hat{\mathbf{g}}_s - \nabla f_\delta(\bar{\mathbf{x}}_s)\|^2\right\} \quad \text{(E.4)}$$

$$\leq 2\left(\min_{s=0,1,2,\ldots,S-1}\|\nabla f_\delta(\bar{\mathbf{x}}_s)\|^2 + \max_{s=0,1,2,\ldots,S-1}\|\hat{\mathbf{g}}_s - \nabla f_\delta(\bar{\mathbf{x}}_s)\|^2\right).$$

This implies that

$$\|\nabla f_\delta(\bar{\mathbf{x}}_{s^\star})\|^2 \leq 2\|\hat{\mathbf{g}}_{s^\star}\|^2 + 2\|\hat{\mathbf{g}}_{s^\star} - \nabla f_\delta(\bar{\mathbf{x}}_{s^\star})\|^2 \quad \text{(E.5)}$$

$$\overset{\text{Eq. (E.4)}}{\leq} 4\left(\min_{s=0,1,2,\ldots,S-1}\|\nabla f_\delta(\bar{\mathbf{x}}_s)\|^2\right) + 4\left(\max_{s=0,1,2,\ldots,S-1}\|\hat{\mathbf{g}}_s - \nabla f_\delta(\bar{\mathbf{x}}_s)\|^2\right) + 2\|\hat{\mathbf{g}}_{s^\star} - \nabla f_\delta(\bar{\mathbf{x}}_{s^\star})\|^2.$$

The next step is to provide the probabilistic bounds on all the terms in the right-hand side of Eq. (E.5). In particular, for each $s = 0, 1, 2, \ldots, S-1$, we have $\bar{\mathbf{x}}_s$ is an output obtained by calling Algorithm 3 with $\mathbf{x}^0$, $d$, $\delta$, $T$ and $\eta = \frac{1}{10}\sqrt{\frac{\delta(\Delta+\delta G)}{cd^{3/2}G^3 T}}$. Then, Eq. (E.3) in the proof of Theorem 3.5 implies that

$$\mathbb{E}[\|\nabla f_\delta(\bar{\mathbf{x}}_s)\|^2] \leq 20\sqrt{\frac{cd^{3/2}G^3}{T}(G + \frac{\Delta}{\delta})}.$$

Using the Markov's inequality, we have

$$\text{Prob}\left(\|\nabla f_\delta(\bar{\mathbf{x}}_s)\|^2 \geq 40\sqrt{\frac{cd^{3/2}G^3}{T}(G + \frac{\Delta}{\delta})}\right) \leq \frac{1}{2}.$$

Thus, we have

$$\text{Prob}\left(\min_{s=0,1,2,\ldots,S-1}\|\nabla f_\delta(\bar{\mathbf{x}}_s)\|^2 \geq 40\sqrt{\frac{cd^{3/2}G^3}{T}(G + \frac{\Delta}{\delta})}\right) \leq 2^{-S}. \quad \text{(E.6)}$$

Further, for each $s = 0, 1, 2, \ldots, S-1$, we have

$$\hat{\mathbf{g}}_s - \nabla f_\delta(\bar{\mathbf{x}}_s) = \frac{1}{B}\sum_{k=0}^{B-1}(\hat{\mathbf{g}}_s^k - \nabla f_\delta(\bar{\mathbf{x}}_s)).$$

By Lemma E.1, we have $\mathbb{E}[\hat{\mathbf{g}}_s^t|\bar{\mathbf{x}}_s] = \nabla f_\delta(\bar{\mathbf{x}}_s)$ and $\mathbb{E}[\|\hat{\mathbf{g}}_s^t\|^2|\bar{\mathbf{x}}_s] \leq 16\sqrt{2\pi}dG^2$. This implies that

$$\mathbb{E}[\hat{\mathbf{g}}_s^t - \nabla f_\delta(\bar{\mathbf{x}}_s)|\bar{\mathbf{x}}_s] = 0, \quad \mathbb{E}[\|\hat{\mathbf{g}}_s^t - \nabla f_\delta(\bar{\mathbf{x}}_s)\|^2] \leq 16\sqrt{2\pi}dG^2.$$

This together with Proposition D.3 yields that

$$\text{Prob}\left(\|\hat{\mathbf{g}}_s - \nabla f_\delta(\bar{\mathbf{x}}_s)\|^2 \geq \frac{\lambda(16\sqrt{2\pi}dG^2)}{B}\right) = \text{Prob}\left(\left\|\sum_{k=0}^{B-1}(\hat{\mathbf{g}}_s^k - \nabla f_\delta(\bar{\mathbf{x}}_s))\right\|^2 \geq \lambda B(16\sqrt{2\pi}dG^2)\right) \leq \frac{1}{\lambda}.$$

Therefore, we conclude that

$$\text{Prob}\left(\max_{s=0,1,2,\ldots,S-1}\|\hat{\mathbf{g}}_s - \nabla f_\delta(\bar{\mathbf{x}}_s)\|^2 \geq \frac{\lambda(16\sqrt{2\pi}dG^2)}{B}\right) \leq \frac{S}{\lambda}. \quad \text{(E.7)}$$

By the similar argument, we have

$$\text{Prob}(\|\hat{\mathbf{g}}_{s^\star} - \nabla f_\delta(\bar{\mathbf{x}}_{s^\star})\|^2 \geq \frac{\lambda(16\sqrt{2\pi}dG^2)}{B}) \leq \frac{1}{\lambda}. \quad \text{(E.8)}$$

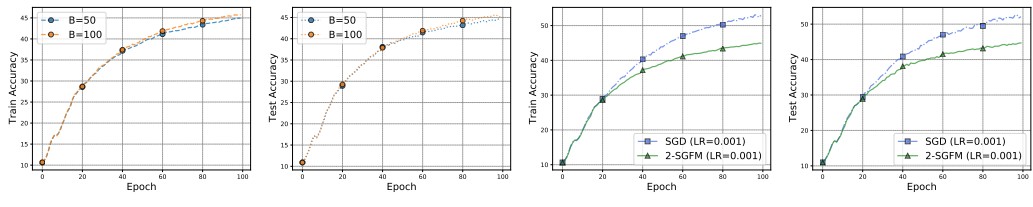

| (a) Effect of $B$ (Train) | (b) Effect of $B$ (Test) | (c) Compare with SGD (Train) | (d) Compare with SGD (Test) |

Figure 5: Addional experimental results on the CIFAR10 dataset [59]. (**a-b**) Performance of 2-SGFM with different choices of $B$. (**c-d**) Performance of 2-SGFM and SGD.

Combining Eq. (E.5), Eq. (E.6), Eq. (E.7) and Eq. (E.8) yields that

$$\text{Prob}\left(\|\nabla f_\delta(\bar{\mathbf{x}}_{s^\star})\|^2 \geq 160\sqrt{\frac{cd^{3/2}G^3}{T}(G+\frac{\Delta}{\delta})} + \frac{\lambda(96\sqrt{2\pi}dG^2)}{B}\right) \leq \frac{S+1}{\lambda} + 2^{-S}, \quad \text{for all } \lambda > 0. \tag{E.9}$$

We set $\lambda = \frac{2(S+1)}{\Lambda}$ and the parameters $(T, S, B)$ as follows,

$$T = cd^{3/2}G^3(G+\tfrac{\Delta}{\delta})(\tfrac{160}{\epsilon^2})^2, \quad S = \lceil\log_2(\tfrac{2}{\Lambda})\rceil, \quad B = \frac{(384\sqrt{2\pi}dG^2)(S+1)}{\Lambda\epsilon^2}.$$

Then, we have

$$\text{Prob}\left(\|\nabla f_\delta(\bar{\mathbf{x}}_{s^\star})\|^2 \geq \epsilon^2\right) \leq \text{Prob}\left(\|\nabla f_\delta(\bar{\mathbf{x}}_{s^\star})\|^2 \geq 160\sqrt{\frac{cd^{3/2}G^3}{T}(G+\frac{\Delta}{\delta})} + \frac{\lambda(96\sqrt{2\pi}dG^2)}{B}\right) \leq \Lambda.$$

By Theorem 3.1, we have $\nabla f_\delta(\bar{\mathbf{x}}_{s^\star}) \in \partial_\delta f(\bar{\mathbf{x}}_{s^\star})$. This together with the above inequality implies that there exists some $T, S, B > 0$ such that the output of Algorithm 4 satisfies that $\mathbb{E}[\min\{\|\mathbf{g}\| : \mathbf{g} \in \partial_\delta f(\bar{\mathbf{x}}_{s^\star})\}] \leq \epsilon$ and the total number of calling the function value oracles is bounded by

$$O\left(d^{\frac{3}{2}}\left(\frac{G^4}{\epsilon^4} + \frac{\Delta G^3}{\delta\epsilon^4}\right)\log_2\left(\frac{1}{\Lambda}\right) + \frac{dG^2}{\Lambda\epsilon^2}\log_2\left(\frac{1}{\Lambda}\right)\right).$$

This completes the proof.

# F  Additional Experimental Results on CIFRA10

We evaluate the performance of our two-phase version of SGFM (Algorithm 4) on the CIFAR10 [59] dataset using convolutional neural networks (CNNs) with ReLU activations. We provide the detailed information about the network architecture as follows,

```python
class CNN_CIFAR(nn.Module):
    def __init__(self):
        super(CNN_CIFAR, self).__init__()
        self.conv1 = nn.Conv2d(3, 6, 5)
        self.conv2 = nn.Conv2d(6, 16, 5)
        self.fc1 = nn.Linear(16*5*5, 120)
        self.fc2 = nn.Linear(120, 84)
        self.fc3 = nn.Linear(84, 10)

    def forward(self, x):
        out = F.relu(self.conv1(x))
        out = F.max_pool2d(out, 2)
        out = F.relu(self.conv2(out))
        out = F.max_pool2d(out, 2)
        out = out.view(out.size(0), -1)
        out = F.relu(self.fc1(out))
        out = F.relu(self.fc2(out))
        out = self.fc3(out)
        out = F.log_softmax(out, dim=1)
        return out
```

Moreover, we summarize the experimental results in Figure 5. In Figure 5a and 5b, we study the effect of batch size $B \geq 1$ in 2-SGFM on the CIFAR10 dataset. In Figure 5c and 5d, We compare the performance of SGD and 2-SGFM. Overall, these results show promising performance of our proposed gradient-free method on solving real-world complex image classification problems.