# OpenReview forum: "Gradient-Free Methods for Deterministic and Stochastic Nonsmooth Nonconvex Optimization"
_NeurIPS.cc/2022/Conference — NeurIPS 2022 Accept_

### Official Review · Reviewer_fAoD · 2022-07-09

**Rating:** 7
**Confidence:** 4
**Soundness:** 3 good
**Presentation:** 3 good
**Contribution:** 3 good

**Summary:**

This paper studied the gradient-free (zeroth-order) methods for the nonsmooth nonconvex optimization problems, and provided the solid theoretical analysis for the proposed gradient-free methods. Notably, it established a useful relationship between the Goldstein subdifferential and uniform smoothing via appeal to the hyperplane separation theorem. Some experimental results demonstrate the effectiveness of the proposed methods.

**Questions:**

Some comments:

1)	Could we further relax the Lipschitz continuous condition in the gradient-free methods for nonsmooth nonconvex optimization ?

2)	In the convergence analysis (Theorem 3.2,3.4-3.6), do we need to choose some small parameter $\delta$ that relies on $d$ and $ \epsilon$, as in the existing zeroth-order methods for smooth optimzation?


**Limitations:**

Yes

**Strengths And Weaknesses:**

Novelty of this paper: Overall, it provides some solid theoretical results on gradient-free methods for the nonsmooth nonconvex optimization.

Weakness of this paper: The Lipschitz continuous condition used in the paper may be not mild.

---

> ### Author Response · Authors · 2022-08-02
> **Reply to Reviewer fAoD**
>
> Thank you for your encouraging comments and positive evaluation! We reply to your questions point-by-point below, and will color all relevant revisions in our paper in ${\color{blue} \textrm{blue}}$.
>
> 1. **The Lipschitz continuous condition used in the paper may be not mild. Could we further relax the Lipschitz continuous condition in the gradient-free methods for nonsmooth nonconvex optimization?**
> We thank the reviewer for pointing this important note out to us. The Lipschitz continuous condition can be relaxed to local Lipschitz continuous condition and all the results still remain the same under minor modification. Indeed, the local Lipschitz condition is enough to guarantee that the randomized smoothing works with sufficiently small \delta. However, it seems that the complete relaxation of the Lipschitz continuous condition is impossible. For gradient-based methods, the Rademacher's theorem (i.e., any Lipschitz function is almost everywhere differentiable) will never hold and we only have the abstract definition of Clarke stationarity (Definition 2.2) without Proposition 2.1. This rules out all the existing theoretical analysis under the randomization scheme. For gradient-free methods, the randomized smoothing will not give a differentiable function with Lipschitz gradients. This also rules out the theoretical analysis in the current manuscript. Nonetheless, there exist finite-time convergent algorithms for some certain classes of non-Lipschitz and nonconvex optimization problems; for example, see the reference: Bian et.al., Complexity analysis of interior point algorithms for non-Lipschitz and nonconvex minimization, Mathematical Programming, 2015.
>
> 2. **In the convergence analysis (Theorem 3.2,3.4-3.6), do we need to choose some small parameter $\delta$ that relies on $d$ and $\epsilon$, as in the existing zeroth-order methods for smooth optimization?**
> Thank you for your helpful comments! We have discussed the difference between the role of $\delta$ in our setting and that in smooth optimization. Indeed, we highlight that $\delta > 0$ is the desired tolerance in our setting. Indeed, $(\delta, \epsilon)$-Goldstein stationarity (see Definition 2.3) relaxes $\epsilon$-Clarke stationarity and our methods pursue an $(\delta, \epsilon)$-stationary point since finding an $\epsilon$-Clarke point is intractable. This is different from smooth optimization where $\epsilon$-Clarke stationarity reduces to $\nabla f(\textbf{x}) \leq \epsilon$ and becomes tractable. In this context, the existing zeroth-order methods are designed to pursue an $\epsilon$-stationary point. Notably, a $(\delta, \epsilon)$-Goldstein stationary point is an $\epsilon$-stationary point in smooth optimization if we choose $\delta$ that relies on $d$ and $\epsilon$. That is why the existing zeroth-order methods set $\delta$ based on $d$ and $\epsilon$ in smooth optimization. Based on the above arguments, we can set the desired tolerance $\delta$ independent of $d$ and $\epsilon$.

---

> > ### Comment · Reviewer_fAoD · 2022-08-07
> > **Reply to the authors' responses**
> >
> > Thanks for the authors' responses.
> >
> > I have read all responses and reviews. The authors have solved my concerns. So I keep my score.

---

### Official Review · Reviewer_VMXu · 2022-07-11

**Rating:** 6
**Confidence:** 2
**Soundness:** 3 good
**Presentation:** 3 good
**Contribution:** 3 good

**Summary:**

This paper established a novel optimality criterion for non-smooth non-convex Lipschitz function called (δ,ε)-Goldstein stationary point and proposed a gradient-free algorithm with its stochastic version by deriving the Goldstein subdifferential and uniform smoothing technical. The last-iterate convergence analysis of the proposed methods was given, which guarantee the convergence to a (δ,ε)-Goldstein stationary point with high probability for both deterministic and stochastic version. State-of-art lower bounds of total oracle calls are given in terms of δ,ϵ and Λ for the proposed methods. Numerical experiments have shown the effectiveness of the proposed methods.

**Questions:**

Though the paper is theoretically sound, there are still some questions need to be discussed in this paper:

1.	The authors proposed a class of subdifferential-based gradient-free algorithms. What is the advantage of the proposed algorithms compared to the gradient-based methods and zeroth-order methods?

2.	The authors compared the performance of the proposed algorithms with different choice on MNIST, which is literally a small-scale and simple dataset. Why not use more larger datasets to verify the excellent performance of the proposed algorithm? In addition, the author's comparison algorithms are too few, which can be compared with some more advanced zeroth-order and gradient-free optimization algorithms like INGD[R1].
[R1] J. Zhang, H. Lin, S. Jegelka, S. Sra, and A. Jadbabaie. Complexity of finding stationary points of nonconvex nonsmooth functions. In ICML, pages 11173–11182. PMLR, 2020.

3.	It seems that the authors trained a quite simple convolutional neural network model on image classification task rather than some more modern and efficient models like [R2] and [R3] according to the experiment results, which is insufficient for validating the effectiveness of the proposed methods.
[R2] He, K., Zhang, X., Ren, S., and Sun, J. Deep residual learning for image recognition. In Proceedings of the IEEE conference on computer vision and pattern recognition, pp. 770–778, 2016

[R3] G. Huang, Z. Liu, L. Van Der Maaten and K. Q. Weinberger, "Densely Connected Convolutional Networks," 2017 IEEE Conference on Computer Vision and Pattern Recognition (CVPR), 2017

4.	Grammar mistakes. Line 342, “We have proposed and analyzed a class of…” “have proposed and analyzed” should be replaced by “propose and analyze”.


**Ethics Review Area:**

["I don’t know"]

**Limitations:**

The authors proposed a class of subdifferential-based gradient-free algorithms. What is the advantage of the proposed algorithms compared to the gradient-based methods and zeroth-order methods? The authors compared the performance of the proposed algorithms with different choice on MNIST, which is literally a small-scale and simple dataset. Why not use more larger datasets to verify the excellent performance of the proposed algorithm?



----------after feedback---------------

The authors address my comments well, so I increased my score.

**Strengths And Weaknesses:**

The theoretical result in this paper is interesting.  The authors established a novel optimality criterion for non-smooth non-convex Lipschitz function called (δ,ε)-Goldstein stationary point and proposed a gradient-free algorithm with its stochastic version by deriving the Goldstein subdifferential and uniform smoothing technical.

---

> ### Author Response · Authors · 2022-08-02
> **Reply to Reviewer VMXu**
>
> Thank you for your time and your input. We reply to your main questions point-by-point below, and we have colored all relevant revisions in our paper in ${\color{blue} \textrm{blue}}$.
>
> 1. **The authors proposed a class of subdifferential-based gradient-free algorithms. What is the advantage of the proposed algorithms compared to the gradient-based methods and zeroth-order methods?**
> We sincerely apologize for the confusion about terminology. **Our algorithms belong to the class of zeroth-order methods**, in the sense that only (noisy) function value is available at each point but gradient information (i.e., subdifferential information) is not available. Note that the optimality notion (i.e., Goldstein stationarity) is indeed defined based on the subdifferential of the function. **However, this does not mean that our algorithms will use the subdifferential information in their scheme**. To our knowledge, our algorithms are **one of the first set of zeroth-order methods** to handle nonsmooth nonconvex optimization with Lipschitz objective function.
>
>      “Gradient-based” methods are usually applied where gradient information is available at each point. Compared to “gradient-based” methods, our methods achieve the finite-time convergence to an approximate Goldstein stationary point even when we only have a function-valued oracle (or a zeroth-order oracle).
>
>      A major reason that we consider the construction of gradient-free methods instead of gradient-based methods is that: **the gradient information sometimes is not readily available for application problems where we only have access to a noisy function value at each point**. This lack of gradient information is a common issue in the context of simulation optimization (Nelson, 2010) and (Hong et al., 2015), where the objective function value is often achieved as the output of a black-box or complex simulator, for which the simulator does not have the infrastructure needed to effectively evaluate gradients; we will also refer to (Ghadimi and Lan, 2013) and (Nesterov and Spokoiny, 2017) for comments on the lack of gradient evaluation in practice.
>
> 2. **The authors compared the performance of the proposed algorithms with different choices on MNIST, which is literally a small-scale and simple dataset. Why not use larger datasets to verify the excellent performance of the proposed algorithm?**
> Thank you for your helpful comments! We agree and have added the experiment with a larger CIFAR10 dataset in the revision; please see Appendix G in the revision. Due to time constraints, we only compare our 2-SGFM with SGD and the numerical results demonstrate the similar performance on MNIST dataset. We will add other methods in the final revision.
>
> 3. **The author's comparison algorithms are too few, which can be compared with some more advanced zeroth-order and gradient-free optimization algorithms like INGD from J. Zhang, H. Lin, S. Jegelka, S. Sra, and A. Jadbabaie. Complexity of finding stationary points of nonconvex nonsmooth functions. In ICML, pages 11173–11182. PMLR, 2020.**
> Thank you for your helpful comments! We have added the INGD in the revision; please see Figure 1 in the page 9 of main context. The numerical result is consistent with Figure 1 in Zhang et. al. INGD outperforms SGD and is competitive with ADAM and AdaGrad. However, we hope to emphasize that INGD is the gradient-based optimization algorithm and requires gradient information at each point. This is in contrast to our method, which is the gradient-free (or zeroth-order) method.

---

> ### Author Response · Authors · 2022-08-02
> **Reply to Reviewer VMXu (continued)**
>
> 4. **It seems that the authors trained a quite simple convolutional neural network model on image classification task rather than some more modern and efficient models like [R2] and [R3] according to the experiment results, which is insufficient for validating the effectiveness of the proposed methods.**
>
>      **[R2] He, K., Zhang, X., Ren, S., and Sun, J. Deep residual learning for image recognition. CVPR, 2016**
>
>      **[R3] G. Huang, Z. Liu, L. Van Der Maaten and K. Q. Weinberger, "Densely connected convolutional networks," CVPR, 2017**
>
>      We would like to thank the reviewer for the helpful comments for potential improvement on the experimental section of the paper. We agree that the numerical results in the previous draft are inadequate to make the point of the paper, and we have conducted additional experiments to better demonstrate the effectiveness of the proposed scheme. For example, we include INGD and present the numerical results on the larger CIFAR10 dataset. We agree that it would be better to validate the effectiveness of the proposed methods using some more modern and efficient models in [R2] and [R3]. However, the practical implementation of our methods for these models would require much more research and may be beyond the scope of this paper. Thus, we cite these references in the revision and reserve further empirical study as future work.
>
> 5. **Grammar mistakes. Line 342, “We have proposed and analyzed a class of…” “have proposed and analyzed” should be replaced by “propose and analyze”.**
> Thank you for pointing it out to us. We have fixed it in the revision.
>
> Thanks again for your remarks! We hope and trust that our replies have alleviated your concerns regarding the merits of our submission, and we look forward to an open-minded discussion if any such concerns remain.

---

### Official Review · Reviewer_xQvN · 2022-07-14

**Rating:** 7
**Confidence:** 4
**Soundness:** 3 good
**Presentation:** 3 good
**Contribution:** 3 good

**Summary:**

This paper introduced two zero-order algorithms to compute the Goldstein stationary points for nonconvex nonsmooth problem. In contrast to the first-order case, the dependence on dimension is unavoidable for algorithms that only use function values. The authors show the gradient of a randomized smoothed function with a delta-ball is belong to the Goldstein delta-subdifferential, which forms the basis for the new gradient-free algorithms. They show the new algorithms compute a Goldstein stationary point in expectation within polynomial oracle complexity and the dimension dependence is only sqrt(d) worse than the convex/smooth case. They also proved a high-probability bound with a two-phase scheme.


**Questions:**

See main comments above.

**Limitations:**

Yes.

**Strengths And Weaknesses:**

As nonconvex nonsmooth problems are everywhere especially in the DL setting, new practical algorithm with finite time oracle complexity is important and desirable nowadays. This paper studies the computation of Goldstein approximate stationary point, which has exhibited attractive algorithmic consequence in recent years. The main contributions are two zero-order finite-time methods which are further built upon an interesting observation that the randomized smoothed function with a delta-ball is belong to the Goldstein delta-subdifferential. The paper is well-written and easy to follow. My comments are as follows:

* Proposition 2.3 basically repeats [78, Lemma 8]. In the proof, it might miss a norm in L629 and L634.
* The step size of Algorithm 1,2,3 seems dependent on the \Delta, which is usually unknown in practice and is not necessary in the first-order setting for Lipschitz functions, e.g., in [79].
* The following reference computing Goldstein stationary points concurrent partly to [30] might be relevant:

[R] Lai Tian, Kaiwen Zhou, and Anthony Man-Cho So. On the finite-time complexity and practical computation of approximate stationarity concepts of Lipschitz functions. ICML, 2022.

* L45: "Clark" -> "Clarke"

---

> ### Author Response · Authors · 2022-08-02
> **Reply to Reviewer xQvN**
>
> Thank you for your encouraging comments and positive evaluation! We reply to your questions point-by-point below, and will color all relevant revisions in our paper in ${\color{blue} \textrm{blue}}$.
>
> 1. **Proposition 2.3 basically repeats [78, Lemma 8]. In the proof, it might miss a norm in L629 and L634.**
> Thank you for pointing this out to us. We have emphasized that Proposition 2.3 is a restatement of [78, Lemma 8] and fixed the missing norm in the revision.
>
> 2. **The step size of Algorithm 1,2,3 seems dependent on the $\Delta$, which is usually unknown in practice and is not necessary in the first-order setting for Lipschitz functions, e.g., in [79].**
> We thank the reviewer for this helpful comment. Indeed as the reviewer pointed out, this $\Delta$ is generally not necessary in the first-order setting for Lipschitz functions. In contrast, for zeroth-order settings, it seems that the information about $\Delta$ is necessary for setting a step size and proving tight theoretical guarantees. Not knowing $\Delta$ may lead to worse complexity for the algorithm, compared to knowing $\Delta$.
>
>       Specifically, the choice of the step size of Algorithm 1,2,3 is crucial to derive the desired main results on complexity; see the key inequalities in Line 717 and 777 from the appendix. If we use a naive step size rule without the prior knowledge of $\Delta$, the proved results may not hold. For example, if we set $\eta = \frac{\delta}{10L}\sqrt{\frac{1}{cd^{3/2}T}}$, the complexity bound will become worse, i.e., $O\left(d^{\frac{3}{2}}\left(\frac{L^2(\Delta + \delta L)^2}{\delta^2\epsilon^4} + \frac{L^4}{\epsilon^4}\right)\right)$. Such a phenomenon is in contrast to the first-order setting where [79] developed an algorithm based on Goldstein subgradient method and set the step size as $\delta$.
>
>      A bit on the **positive** side is that, technically the zeroth-order setting does not need exact knowledge of $\Delta$, which as the reviewer pointed out is not available in practice. Knowing an estimate of $\Theta(\Delta)$ would suffice for the algorithm design and theoretical results to go through. Of course, if the provided estimation of $\Delta$ is too loose, the resulting complexity can be adversely affected but up to a constant. In practice, we would recommend users to identify an estimation of \Delta given the specific problem setting, but indeed the complexity results can comprise for the zero-order setting if no such information about $\Delta$ is available.
>
>      Intuitively, the first-order information gives more information than zeroth order information such that the step size can be independent of more problem parameters while not sacrificing the complexity results. In the revision, we have highlighted this point and remarked that the design of a more practical step size rule is a promising future direction.
>
> 3. **The following reference computing Goldstein stationary points concurrent partly to [30] might be relevant: L. Tian, K. Zhou, and A. M-C. So. On the finite-time complexity and practical computation of approximate stationarity concepts of Lipschitz functions. ICML, 2022.**
> Thank you for pointing this relevant reference to us. We have added adequate discussion about the reference in the revision.
>
> 4. **"Clark" -> "Clarke".**
> Thank you for pointing this typo out to us. We have fixed it in the revision.

---

### Meta-Review · Area_Chair_84eU · 2022-08-30

**Recommendation:** Accept
**Confidence:** Certain

**Metareview:**

The authors introduce two derivative-free algorithms for computing the Goldstein stationary points in the context of nonconvex nonsmooth optimization, and show that they enjoy polynomial complexity (in expectation), while the dimension dependence (which is unavoidable in the derivative-free setting) is worse by only an sqrt(d) factor compared to the convex/smooth case. A high-probability bound with a two-phase scheme was established as well.

The reviewers described the strengths of the paper in the following way:
- The paper is well-written and easy to follow.
- The theoretical result in this paper is interesting.
- The authors established a novel optimality criterion for non-smooth non-convex Lipschitz function called (δ,ε)-Goldstein stationary point and proposed a gradient-free algorithm with its stochastic version by deriving the Goldstein subdifferential and uniform smoothing technical.
- Overall, it provides some solid theoretical results on gradient-free methods for the nonsmooth nonconvex optimization.

Some criticism was raised, but the authors managed to address it in their rebuttal. I agree with the collective judgment of the reviewers that this paper clearly passes the bar of acceptance. Please make sure that all criticism is properly addressed in the camera-ready version of the paper as well.

Congratulations on a nice paper!

AC

**Award:**

No

---

### Decision · Program_Chairs · 2022-09-14

Accept